# Natural variation in a type-A response regulator confers maize chilling tolerance

Rong Zeng[1,2], Zhuoyang Li[1,2], Yiting Shi[1], Diyi Fu[1], Pan Yin[1], Jinkui Cheng[1], Caifu Jiang[1] & Shuhua Yang 🆔 [1✉]

Maize (*Zea mays* L.) is a cold-sensitive species that often faces chilling stress, which adversely affects growth and reproduction. However, the genetic basis of low-temperature adaptation in maize remains unclear. Here, we demonstrate that natural variation in the type-A *Response Regulator 1* (*ZmRR1*) gene leads to differences in chilling tolerance among maize inbred lines. Association analysis reveals that InDel-35 of *ZmRR1*, encoding a protein harboring a mitogen-activated protein kinase (MPK) phosphorylation residue, is strongly associated with chilling tolerance. ZmMPK8, a negative regulator of chilling tolerance, interacts with and phosphorylates ZmRR1 at Ser15. The deletion of a 45-bp region of ZmRR1 harboring Ser15 inhibits its degradation via the 26 S proteasome pathway by preventing its phosphorylation by ZmMPK8. Transcriptome analysis indicates that ZmRR1 positively regulates the expression of *ZmDREB1* and *Cellulose synthase* (*CesA*) genes to enhance chilling tolerance. Our findings thus provide a potential genetic resource for improving chilling tolerance in maize.

---

[1] State Key Laboratory of Plant Physiology and Biochemistry, College of Biological Sciences, Center for Crop Functional Genomics and Molecular Breeding, China Agricultural University, Beijing, China. [2] These authors contributed equally: Rong Zeng, Zhuoyang Li. ✉email: yangshuhua@cau.edu.cn

Chilling tolerance is a key agronomic trait of crops grown in temperate regions. The selection of low-temperature-tolerant crops is crucial for adapting agricultural practices to changing climate conditions. Maize (*Zea mays* L.), a critically important food resource, originated in tropical regions and is inherently sensitive to low temperatures, which limits its growth at higher latitudes[1,2]. Low temperatures severely affect maize's early vigor and productivity[3]. The chilling tolerance of maize must be improved by developing and selecting chilling-tolerant genotypes using breeding and genomic approaches.

Chilling stress causes reduced photosynthetic efficiency, membrane rigidification, and altered reactive oxygen species levels in plant cells[4]. Traditional genetic and molecular analyses have been used to mine major QTLs/genes that control cold tolerance in rice (*Oryza sativa*), including cold tolerance gene *Ctb1*, *Low-temperature germinability on chromosome 3* (*qLTG3*), *Chilling tolerance divergence1* (*COLD1*), *Cold tolerance at booting stage 4a* (*CTB4a*), *bZIP73*, *Low temperature growth 1* (*LTG1*), and *HAN1*[5–11]. Genomics-assisted approaches have helped uncover the single nucleotide polymorphisms (SNPs) associated with these major QTLs/genes, which underpin the differences between alleles and provide molecular markers for developing new chilling-tolerant varieties. Linkage mapping and association mapping have predicted candidate QTLs/genes related to photosynthesis, sugar metabolism, and secondary metabolism associated with low-temperature adaptation in maize[12–15]. A deeper understanding of genetic variations underlying chilling tolerance in maize will facilitate the development of superior, cold-adapted maize varieties.

To survive in cold temperatures, plants must perceive the cold signal and transduce it to activate *cold-responsive* (*COR*) gene expression[4,16]. Dehydration-responsive element (DRE) binding factor 1 s (DREB1s), also known as C-repeat binding factors (CBFs), play a central role in plant response to cold stress in plants[4,17]. The activation of cold signal transduction and downstream responsive events relies on phosphorylation mediated by protein kinases[18]. The protein kinase SnRK2.6 (Open stomata 1 (OST1)/SNF1-related protein kinase 2.6) phosphorylates ICE1 (Inducer of CBF expression 1), which activates *DREB1* gene transcription and triggers the cold stress response in *Arabidopsis thaliana*[19–21]. Cold-responsive protein kinase 1 (CRPK1) phosphorylates 14-3-3 proteins, which import from the cytoplasm into the nucleus and destabilize DREB1 proteins to negatively regulate *Arabidopsis* freezing tolerance[22]. The mitogen-activated protein kinases MPK3/6 regulate the cold response by phosphorylating and modulating the stability of ICE1[23,24]. The MEKK1-MEK2-MPK4 cascade positively regulates *DREB1* gene expression and freezing tolerance by antagonizing the MKK4/5-MPK3/6 pathway in *Arabidopsis*[24,25]. In rice, OsMPK3 positively regulates chilling tolerance by inhibiting OsICE1 degradation under cold stress[26]. Heterologous expression of maize *ZmMKK1* or *ZmMKK4* increases cold tolerance in tobacco (*Nicotiana tabacum*) or *Arabidopsis*[27,28]. Similarly, the ectopic expression of the maize MAP kinase gene *ZmSIMK1* enhanced tolerance to salt and drought stress[29,30].

Cytokinin signal transduction is mediated by a conserved two-component system involving a phosphorelay from histine kinase receptors and histidine containing phosphotransfer proteins to downstream response regulators (RRs)[31]. Response regulators majorly consist of type-A and type-B RRs. Type-B RRs are transcription factors that can directly promote the expression of type-A *RR* genes. Type-A RRs, which contain a receiver domain but lack of output domain, are responsible for repressing cytokinin signaling via a negative feedback loop[32]. It has been shown that type-A RRs are extensively involved in abiotic stress responses, such as drought and cold stress responses. SnRK2s

phosphorylate the Ser residue of ARR5 to enhance its stability and plant drought tolerance[33]. Cold induces the expression of *ARR5*, *6*, *7*, and *15* genes to enhance the freezing tolerance in *Arabidopsis*[34,35]. In rice, *OsRR6* is induced by various abiotic stresses and enhances drought and salinity tolerance, while *OsRR9* and *OsRR10* negatively regulate salinity tolerance in rice[36,37]. Nevertheless, the mechanism by which type-A RRs regulate cold stress and type-A RRs is regulated under cold stress in crops remain elusive.

Here, we demonstrate that the type-A RR ZmRR1 and mitogen-activated protein kinase ZmMPK8 are positive and negative regulators of maize chilling tolerance. In addition, ZmRR1 is phosphorylated at Ser15 by ZmMPK8 during cold treatment. A natural variation of ZmRR1 lacking a 15-amino-acid region harboring Ser15 cannot be phosphorylated by ZmMPK8, resulting in enhanced ZmRR1 protein stability and chilling tolerance. These findings provide an in-depth understanding of chilling stress tolerance in maize and reveal an important genetic target for breeding chilling-tolerant maize varieties.

## Results

**ZmRR1 positively regulates maize chilling tolerance**. To identify novel components involved in chilling tolerance in maize, we screened a population of transgenic maize plants overexpressing >700 maize genes in inbred line LH244 background (generated by the Center for Crop Functional Genomics and Molecular Breeding, CAU) under cold treatment and identified *ZmRR1* (*Zea mays Response Regulator 1*), which encodes a type-A response regulator. Overexpressing *ZmRR1* significantly enhanced chilling tolerance in maize seedlings (Fig. 1a, Supplementary Fig. 1a, b). Under cold treatment at 4 °C, the relative injured area in ZmRR1-overexpression lines was smaller than that in wild-type LH244 (Fig. 1b). Ion leakage assays indicated that plasma membrane damage caused by chilling stress was greatly reduced in ZmRR1-overexpression lines vs. the wild type (Fig. 1c).

We generated the *ZmRR1* mutants, *zmrr1-c1*, and *zmrr-c2*, by CRISPR (Clustered Regularly Interspaced Short Palindromic Repeats)/Cas9-mediated gene editing. These mutants carried a 74-bp deletion (from 47 to 120 bp after the ATG start codon) and 1 bp insertion (at 408 bp after ATG) in *ZmRR1*, respectively (Supplementary Fig. 1c), which led to frameshifts. The *zmrr1-c1* and *zmrr1-c2* mutants were more sensitive to chilling stress than the wild type, showing a larger relative injured area and greater ion leakage (Fig. 1d–f). We also crossed *zmrr1-c1* with *zmrr1-c2*, and F1 progeny plants were comparable to the single mutants of *zmrr1-c1* and *zmrr-c2* in terms of chilling sensitivity (Supplementary Fig. 1d, e). These results demonstrate that *ZmRR1* plays a positive role in regulating chilling tolerance.

**Identification of a favorable allele of *ZmRR1* with enhanced chilling tolerance**. The accumulation of the type-A RR ZmRR1 was rapidly induced by cytokinin at both transcriptional[38] and protein levels (Supplementary Fig. 2a, b), we therefore examined whether the abundance of ZmRR1 is regulated by cold stress. *ZmRR1* transcript levels decreased slightly in the wild-type plants after 6 h of cold treatment (Supplementary Fig. 2c), whereas ZmRR1 protein levels (as detected using the anti-ZmRR1 antibody) was increased in these plants after cold treatments compared to the plants grown at 25 °C (Fig. 1g and Supplementary Fig. 3). This finding indicates that cold stress positively regulates ZmRR1 protein levels in maize.

Then we resequenced the *ZmRR1* from 161 maize inbred lines collected from tropical and temperate regions. We identified 20 SNPs (which cause nonsynonymous mutation) and 3 insertion/deletions (InDels) with minor allele frequency (MAF) >5%. To

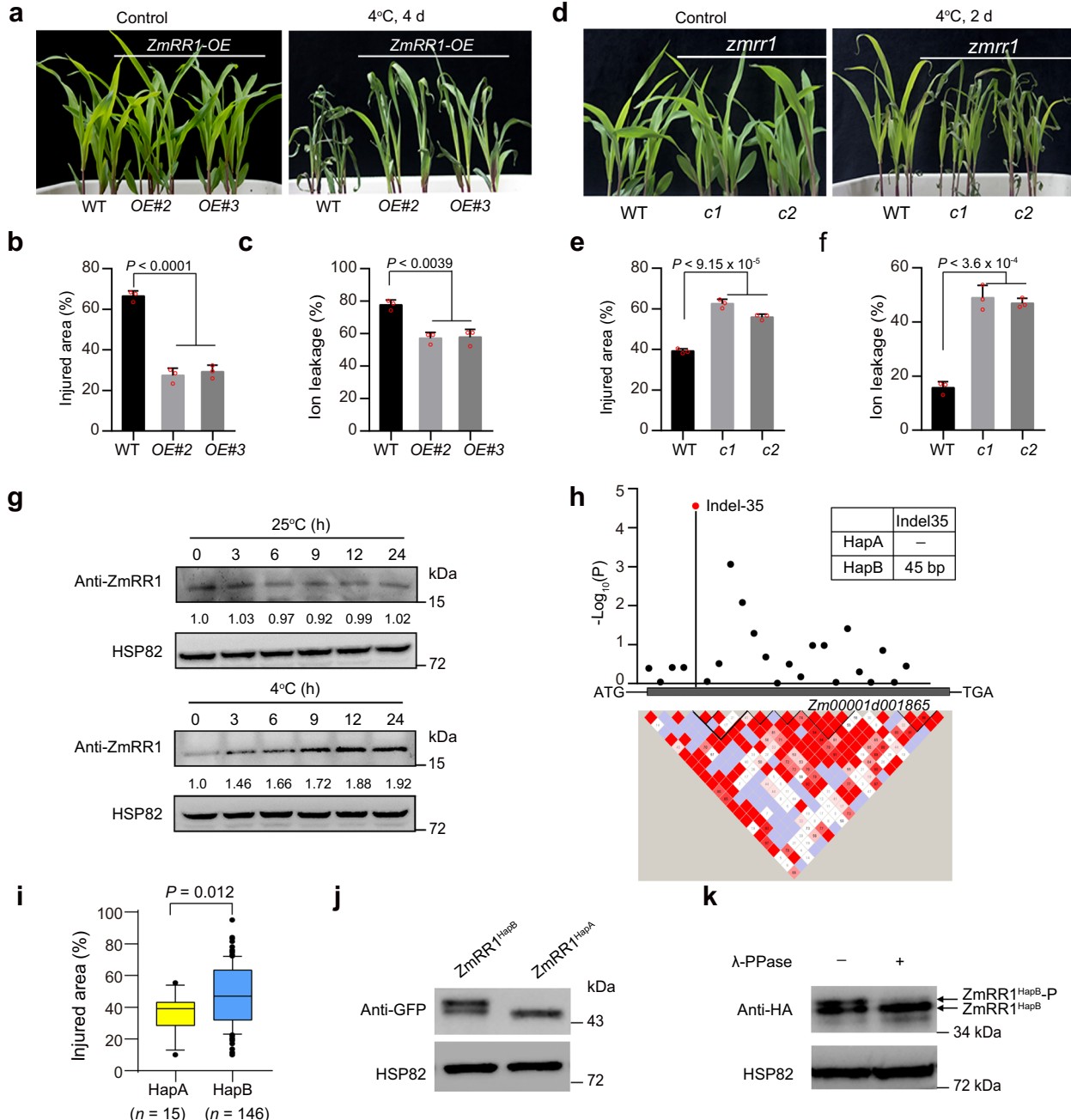

**Fig. 1 Natural variation in the *ZmRR1* coding region confers chilling tolerance in maize. a–c** Chilling phenotypes (**a**), injured area (**b**), and ion leakage (**c**) of *ZmRR1*-overexpressing transgenic plants (*OE#2*, *OE#3*) after cold treatment. 14-day-old seedlings grown at 25 °C were incubated at 4 °C for 4 d. Representative images were taken after 2 days of recovery. **d–f** Chilling phenotypes (**d**), injured area (**e**), and ion leakage (**f**) of *zmrr1* mutants (*c1* and *c2*) under cold conditions. **g** Cold induces ZmRR1 protein accumulation. Ten-day-old WT seedlings were incubated at 25 °C or 4 °C for the indicated times. ZmRR1 was detected with anti-ZmRR1 antibody. HSP82 was used as a control. **h** ZmRR1-based association mapping and pairwise LD analysis. Indel-35 ($P = 2.75 \times 10^{-5}$) is highlighted by a red triangle. Blackdots represent nonsynonymous variants in the coding region of *ZmRR1*. Haplotypes of *ZmRR1* were grouped according to the significant variant (Indel-35). **i** Boxplot showing the injured area of each haplotype. The box shows the median, and the lower and upper quartiles, and the dots denote outliers. ($n = 15$ for HapA, $n = 146$ for HapB). Statistical significance was determined by a two-sided *t*-test. **j** Immunoblot analysis of ZmRR1 proteins in maize protoplasts expressing ZmRR1$^{HapA}$-GFP and ZmRR1$^{HapB}$-GFP. ZmRR1 was detected with anti-GFP antibody. HSP82 was used as a control. **k** Immunoblot analysis of ZmRR1 proteins in maize protoplasts expressing HF-ZmRR1$^{HapB}$ with or without λ-phosphatase treatment. ZmRR1 was detected with anti-HA antibody. HSP82 was used as a control. In **b**, **c**, **e**, **f**, each bar represents the mean ± SD of three independent experiments. The statistical significance was determined by a two-sided *t*-test. In **g**, **j**, **k**, a representative experiment from three independent experiments is shown. Source data underlying Fig. 1b, c, e-g, and i-k are provided as a Source Data file.

investigate whether natural variations in *ZmRR1* contribute to the variation in chilling tolerance in maize, we performed association analysis all these SNPs/InDels with chilling tolerance phenotypes using TASSEL (Bonferroni threshold $P < 4.3 \times 10^{-3}$). InDel35 was the most highly associated genetic polymorphism with chilling tolerance ($P = 2.75 \times 10^{-5}$; Fig. 1h). The 161 maize varieties were classified into two haplotype groups (HapA and HapB) based on the haplotypes of InDel35 (Fig. 1h). The HapA and HapB groups contained 15 and 146 inbred lines, respectively (Supplementary Data 1). Since wild-type inbred line LH244 used in this study belongs to HapB, ZmRR1 is referred to as ZmRR1$^{HapB}$ hereafter. HapB members had a significantly higher injured area ratio than HapA members ($P = 0.012$) (Fig. 1i). Thus, we designated HapA and HapB as resistant and susceptible alleles of *ZmRR1*, respectively.

To dissect the function of ZmRR1$^{HapA}$ and ZmRR1$^{HapB}$ in regulating plant cold tolerance, we overexpressed them in *Arabidopsis* and tested their freezing tolerance. Transgenic plants overexpressing both *ZmRR1$^{HapA}$* and *ZmRR1$^{HapB}$* displayed higher freezing tolerance than the wild-type plants. *ZmRR1$^{HapA}$* overexpression seedlings showed even higher survival rate and lower ion leakage than *ZmRR1$^{HapB}$* overexpression lines (Supplementary Fig. 4). These results further support the notion that *ZmRR1$^{HapA}$* is an elite allelic variation for improving plant cold tolerance.

To examine how the natural variation in the *ZmRR1* coding region affects its biological function, we compared ZmRR1 protein patterns in maize protoplasts expressing *ZmRR1$^{HapA}$-GFP* (amplified from inbred line B111) and *ZmRR1$^{HapB}$-GFP* (amplified from inbred line LH244) driven by a constitutive Super promoter. Intriguingly, two bands corresponding to ZmRR1$^{HapB}$-GFP were detected, whereas ZmRR1$^{HapA}$-GFP only showed one band (Fig. 1j). The upper band of HA-Flag-ZmRR1$^{HapB}$ (HF-ZmRR1$^{HapB}$) disappeared after $\lambda$ alkaline phosphatase treatment (Fig. 1k), suggesting that ZmRR1$^{HapB}$ could be phosphorylated and this phosphorylation does not occur in the ZmRR1$^{HapA}$ variant.

**ZmRR1 is phosphorylates by ZmMPK8**. By analyzing the amino acid sequences of ZmRR1$^{HapA}$ and ZmRR1$^{HapB}$, we observed a conserved serine-proline (SP) motif within the most highly associated InDel35 in ZmRR1$^{HapB}$ (Supplementary Fig. 3c), which is predicted to be a potential mitogen-activated protein kinase (MAPK) phosphorylation motif[39,40]. We performed a yeast two-hybrid assay and found that full-length ZmRR1$^{HapB}$ interacted with ZmMPK2 and ZmMPK8, but not with ZmMPK4 or ZmMPK5, in yeast cells (Fig. 2a). ZmMPK2 and ZmMPK8 are clade C MPKs. Plants overexpressing both *ZmMPK2* and *ZmMPK8* genes showed similar defective cold response (described in detail below). Therefore, we focused our analysis on ZmMPK8. Deletion analysis showed that ZmRR1$^{HapB}$ interacted with the kinase domain of ZmMPK8 (171−369 aa; ZmMPK8-KD), but not with the truncated form of ZmMPK8 without kinase domain (1−170 aa; ZmMPK8-N) (Fig. 2b).

ZmRR1$^{HapB}$ protein localized to the cytosol of maize protoplasts expressing *ZmRR1$^{HapB}$-GFP* (Supplementary Fig. 5), which is consistent with previous findings[41]. ZmMPK8 protein localized to both cytosol and nucleus in maize protoplasts (Supplementary Fig. 5). In a bimolecular fluorescence complementation (BiFC) assay, full-length ZmMPK8 interacted with ZmRR1$^{HapB}$ in the cytosol, whereas ZmMPK8-N (as a control) failed to interact with ZmRR1 (Fig. 2c). These data suggest that the kinase domain of ZmMPK8 is required for their interaction.

To examine the effect of natural variations of ZmRR1 on its binding affinity with ZmMPK8, we expressed ZmRR1$^{HapA}$ and ZmRR1$^{HapB}$ in yeast cells and perform a yeast two-hybrid assay. The interaction of ZmRR1$^{HapA}$ with ZmMPK8 in yeast was weaker than its interaction with ZmRR1$^{HapB}$ (Fig. 2b). To validate the interaction between ZmRR1 with ZmMPK8, we performed a glutathione S-transferase (GST) pull-down assay. ZmRR1$^{HapA}$ and ZmRR1$^{HapB}$ fused to GST-tag that were expressed and purified from *E. coli* were incubated with His-tagged ZmMPK8 protein. Much more GST-ZmRR1$^{HapB}$ was pulled down by His-ZmMPK8 than ZmRR1$^{HapA}$ (Fig. 2d), indicating that the binding affinity of ZmMPK8 to ZmRR1$^{HapB}$ is higher than that to ZmRR1$^{HapA}$ in vitro.

We further verified the interaction of ZmRR1 variants and ZmMPK8 by performing co-immunoprecipitation (co-IP) assays in maize protoplasts expressing *ZmRR1$^{HapB}$-GFP* and *ZmMPK8-MYC* or *ZmRR1$^{HapA}$-GFP* and *ZmMPK8-MYC*. ZmRR1$^{HapB}$-GFP proteins were immunoprecipitated with anti-GFP antibody, and ZmMPK8 was detected with the anti-MYC antibody. Both ZmRR1$^{HapA}$ and ZmRR1$^{HapB}$ were associated with ZmMPK8. However, the association between ZmRR1$^{HapA}$ and ZmMPK8 was less than half of ZmRR1$^{HapB}$ and ZmMPK8 (Fig. 2e). Together, these results indicate that both ZmRR1$^{HapA}$ and ZmRR1$^{HapB}$ interact with ZmMPK8 in vitro and in vivo, and that ZmRR1$^{HapB}$ has higher affinity for ZmMPK8 than ZmRR1$^{HapA}$.

**ZmMPK8 negatively regulates chilling tolerance**. To explore the role of ZmMPK8 in regulating maize chilling tolerance, we generated transgenic maize plants overexpressing *ZmMPK8* driven by the ubiquitin promoter (Supplementary Fig. 6a) and subjected them to a chilling tolerance assay. Two independent *ZmMPK8*-overexpression lines (*ZmMPK8-OE#10* and *#13*) displayed substantially reduced chilling tolerance compared to the wild-type LH244 (Fig. 3a), which showed increased relative injury area and ion leakage after chilling treatment (Fig. 3b, c). Similarly, transgenic maize plants overexpressing *ZmMPK2* also showed reduced chilling tolerance (Supplementary Fig. 6b–e), suggesting that ZmMPK2 shares a similar function with ZmMPK8 in regulating maize chilling tolerance.

We created two mutants of *ZmMPK8* named *zmmpk8-c1* and *zmmpk8-c2* via CRISPR/Cas9-mediated gene editing. The *zmmpk8-c1* mutant carried 1-bp insertion at 306 bp downstream of the start codon (ATG), and *zmmpk8-c2* harbored a 11-bp deletion between nucleotides 299 and 309 after the start codon (Supplementary Fig. 7a), which caused frameshifts and early termination of translation after amino acids 102 and 109, respectively. Following incubation at 4 °C, the *zmmpk8* mutants showed dramatically smaller relative injury area and reduced ion leakage than the wild type (Fig. 3d–f). Allelism test of *zmmpk8-c1* and *zmmpk8-c2* mutants showed that F1 progeny seedlings of *zmmpk8-c1* crossed with *zmmpk8-c2* showed comparable chilling tolerance to the single mutants of *zmmpk8-c1* and *zmmpk8-c2* (Supplementary Fig. 7b, c), indicating that the chilling tolerance is caused by the mutations of *ZmMPK8*. These data indicate that ZmMPK8 is a negative regulator of chilling tolerance in maize.

To decipher the genetic relevance of *ZmRR1* and *ZmMPK8* genes, we generated the *zmrr1-c1 zmmpk8-c1* double mutant by crossing *zmrr1-c1* with *zmmpk8-c1*. The *zmmpk8-c1 zmrr1-c1* double mutant showed more chilling sensitivity than the *zmmpk8-c1* single mutant (Fig. 3g). It largely behaved like the *zmrr1-c1* single mutant in terms of relative injured area and ion leakage (Fig. 3g–i). Thus, *ZmMPK8* negatively regulates maize chilling tolerance through modulating ZmRR1.

**ZmMPK8 phosphorylates ZmRR1$^{HapB}$ in vitro and in vivo**. As ZmMPK8 interacts with ZmRR1, we reasoned that ZmRR1$^{HapB}$

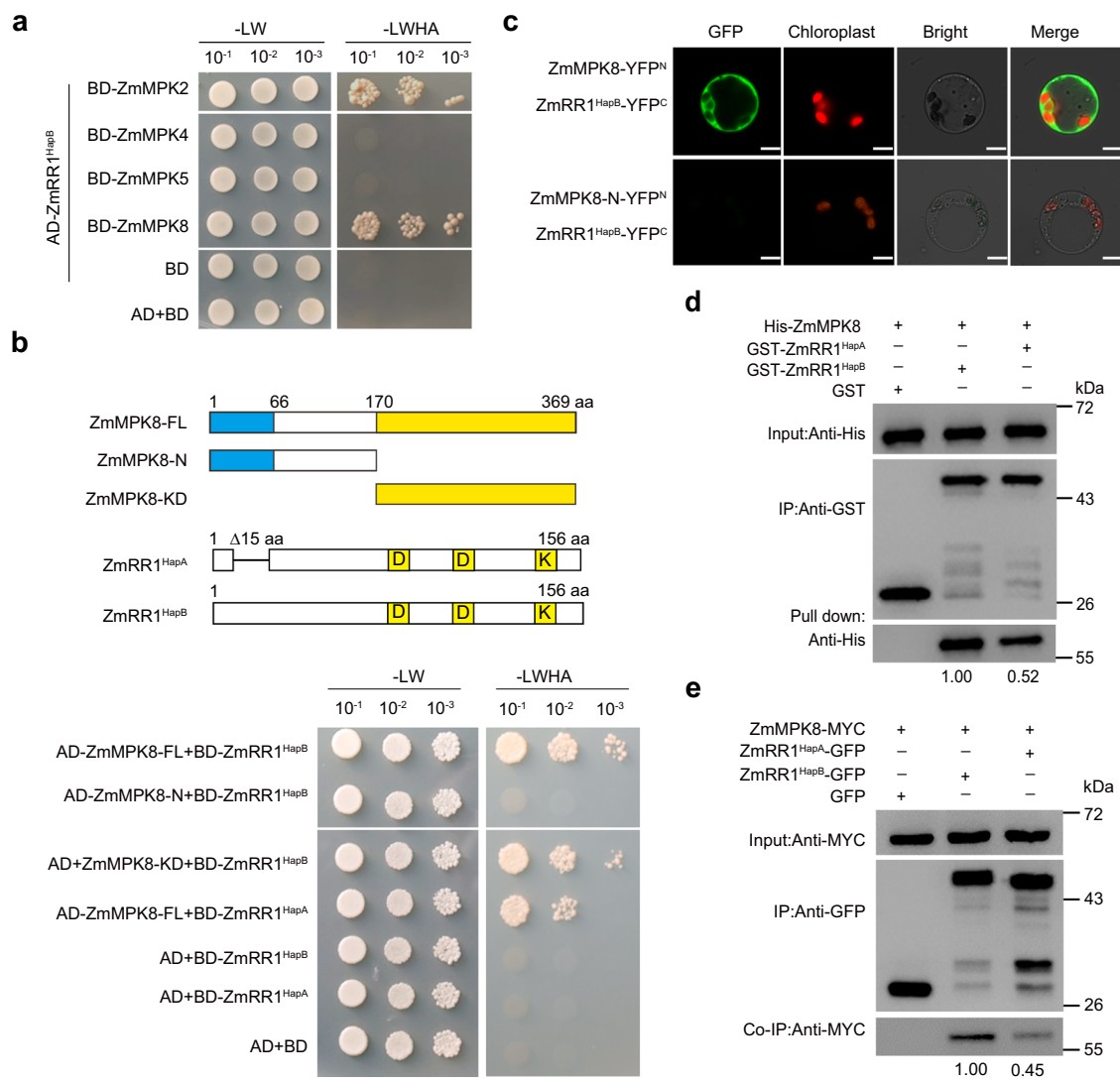

**Fig. 2 ZmMPK8 interacts with ZmRR1 in vitro and in vivo. a** Interactions of ZmMPKs with ZmRR1 in yeast. Yeast cells were grown on SD/-Leu/-Trp or SD/-Leu/-Trp/-His/-Ade medium. **b** The interaction of ZmMPK8 with ZmRR1HapA and ZmRR1HapB in yeast. Diagram of full-length and truncated ZmMPK8 constructs with specific deletions and ZmRR1HapA and ZmRR1HapB constructs (top). **c** Bimolecular fluorescence complementation assay (BiFC) showing the interaction between ZmMPK8 and ZmRR1 in maize protoplasts. Maize protoplasts co-expressing ZmMPK8-YFP$^N$ or ZmMPK8-N-YFP$^N$ with ZmRR1-YFP$^C$ were incubated in the dark for 15 h. The GFP signals were visualized by confocal microscopy. Bar = 5 μm. **d** In vitro pull-down assay showing the interactions between ZmMPK8 and ZmRR1HapA/B. **e** Co-IP showing the interactions between ZmMPK8 and ZmRR1HapA/B. Total proteins from maize protoplasts co-expressing ZmMPK8-MYC with ZmRR1HapA-GFP or ZmRR1HapB-GFP were immunoprecipitated with GFP beads and detected with anti-MYC antibody. In **a–e**, a representative experiment from three independent experiments is shown. Source data underlying Fig. 2d and e are provided as a Source Data file.

(ZmRR1) might be a substrate of ZmMPK8. To test this hypothesis, we performed in vitro phosphorylation assays. By comparing the sequence of ZmMPK8 with that of its homolog AtMPK4, we identified Tyr-113 as an important residue for auto-kinase activity[42] (Supplementary Fig. 8a). We therefore generated purified His-tagged ZmMPK8$^{Y113C}$ (mimic constitutively active form of ZmMPK8) and ZmMPK8 proteins and used them in vitro phosphorylation assays using ZmRR1HapB as a substrate. As expected, ZmMPK8$^{Y113C}$ showed strong auto-phosphorylation activity (Supplementary Fig. 8b). Intriguingly, ZmMPK8 also showed auto-phosphorylation activity, although it was weaker than that of ZmMAPK8$^{Y113C}$ (Supplementary Fig. 8b). Moreover, both of ZmMPK8 and ZmMAPK8$^{Y113C}$ proteins phosphorylated ZmRR1HapB in vitro (Fig. 4a; Supplementary Fig. 7b).

To determine whether ZmMPK8 phosphorylates ZmRR1HapB in planta, we expressed ZmRR1HapB tagged with HA and FLAG (HF-ZmRR1HapB) together with ZmMPK8-MYC or HF-ZmRR1HapB alone in the leaves of *N. benthamiana*. Without co-expression of ZmMPK8, two bands of ZmRR1HapB protein were detected (Fig. 4b), which is consistent with the previous observation (Fig. 1k). However, the relative intensity of the phosphorylated form (upper band) of HF-ZmRR1HapB became much stronger after co-expression of ZmMPK8-MYC (Fig. 4b). Furthermore, we expressed HF-ZmRR1HapB in maize protoplasts derived from the wild-type LH244 and *zmmpk8-c1* plants. Much less HF-ZmRR1HapB phosphorylated form was detected in the *zmmpk8* vs. wild-type background (Fig. 4c), suggesting that ZmMPK8 contributes to the phosphorylation of ZmRR1HapB in planta.

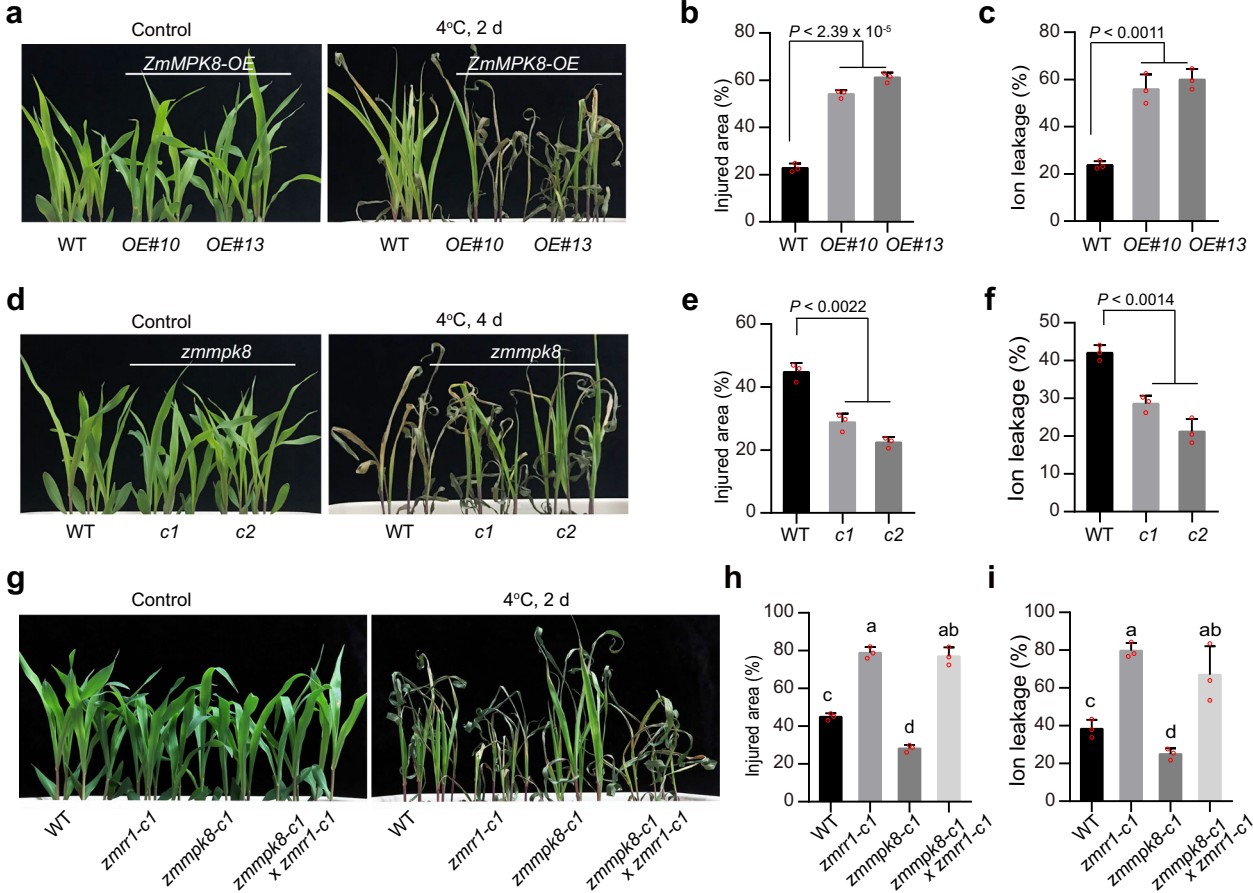

**Fig. 3 ZmMPK8 negatively regulates chilling tolerance. a–c** Chilling phenotypes (**a**), injured area (**b**), and ion leakage (**c**) of *ZmMPK8*-overexpression transgenic plants after cold treatment. **d–f** Chilling phenotypes (**d**), injured area (**e**), and ion leakage (**f**) of *zmmpk8* (*c1* and *c2*) mutants after cold treatment. **g–i** Genetic interaction of *ZmRR1* and *ZmMPK8*. Chilling phenotypes (**g**), injured area (**h**), and ion leakage (**i**) of *zmrr1-c1, zmmpk8-c1,* and *zmrr1 zmmpk8* double mutants after cold treatment were shown. In **b, c, e, f, h, i**, each bar represents the mean ± SD of three independent experiments. In **b, c, e, f**, the statistical significance was determined by a two-sided *t*-test. In **h** and **i**, different letters represent significant differences (*P* < 0.05, one-way ANOVA). Source data underlying Fig. 3b, c, e, f, h, and i are provided as a Source Data file.

Finally, we tested whether ZmMPK8 phosphorylates ZmRR1$^{HapB}$ by performing an in-gel kinase assay using total proteins extracted from the wild type, *zmmpk8-c1*, and *ZmMPK8*-overexpression plants under cold stress using recombinant GST-ZmRR1$^{HapB}$ as a substrate. The kinase activity of ZmMPK8 (visible as signals at ~42 kDa) in wild-type plants was activated after cold treatment (Fig. 4d). The cold-induced kinase activity of ZmMPK8 was much stronger in *ZmMPK8*-overexpression plants compared to the wild type (Fig. 4d). These data demonstrate that ZmMPK8 phosphorylates ZmRR1$^{HapB}$ in vitro and in vivo. However, the kinase activity was much weaker but not fully abolished in the *zmmpk8* mutant vs. the wild type, suggesting that ZmRR1$^{HapB}$ might be also phosphorylated by other ZmMPKs, possibly including ZmMPK2.

To search for potential ZmMPK8 phosphorylation sites in ZmRR1$^{HapB}$, we performed liquid chromatography (LC)-mass spectrometry (MS)/MS of GST-ZmRR1$^{HapB}$ recombinant protein that has been phosphorylated by His-ZmMPK8 in vitro and identified Ser15 as a candidate phosphorylation site in GST-ZmRR1 (Supplementary Fig. 9a). When Ser15 of GST-ZmRR1$^{HapB}$ was mutated to Ala, the mimic nonphosphorylated form, GST-ZmRR1$^{S15A}$, was not phosphorylated by His-ZmMPK8$^{Y113C}$ in vitro (Fig. 4a). Furthermore, we enriched ZmRR1$^{HapB}$ protein from protoplasts transformed with *ZmRR1$^{HapB}$-MYC* by immunoprecipitation using anti-MYC beads and subjected it to LC-MS/MS to pinpoint its

phosphorylation sites in vivo. Consistent with the in vitro result (Supplementary Fig. 9a), the phosphorylation modification of ZmRR1$^{HapB}$ at Ser15 was identified in planta (Supplementary Fig. 9b). Indeed, only the nonphosphorylated band (lower band) was detected when we expressed ZmRR1$^{S15A}$-GFP in maize protoplasts (Fig. 4e), which mimicked the scenario of ZmRR1$^{HapA}$ (Fig. 1j). These results indicate that Ser15 of ZmRR1$^{HapB}$ is the phosphorylation site of ZmMPK8, which is absent in ZmRR1$^{HapA}$.

**ZmMPK8 accelerates the degradation in ZmRR1$^{HapB}$ under cold stress.** To dissect the biological significance of the phosphorylation of ZmRR1$^{HapB}$ by ZmMPK8, we examined the protein abundance of ZmRR1$^{HapB}$ in the wild type, *ZmMPK8-OE*, and *zmmpk8-c1* plants via immunoblot analysis using anti-ZmRR1 antibody. Cold-induced increases in ZmRR1$^{HapB}$ protein levels were slightly attenuated in the *ZmMPK8*-overexpression plants but were dramatically increased in the *zmmpk8-c1* mutant compared to the wild type (Fig. 4f). After we transiently expressed ZmRR1$^{HapB}$-GFP with different amounts of ZmMPK8-MYC in maize protoplasts, ZmRR1$^{HapB}$ protein levels substantially decreased with increasing levels of ZmMPK8 (Fig. 4g). This decrease in ZmRR1$^{HapB}$ protein abundance was fully inhibited by treatment with the 26 S proteasome inhibitor MG132 (Fig. 4g). By contrast, the protein levels of ZmRR1$^{S15A}$-GFP were not affected

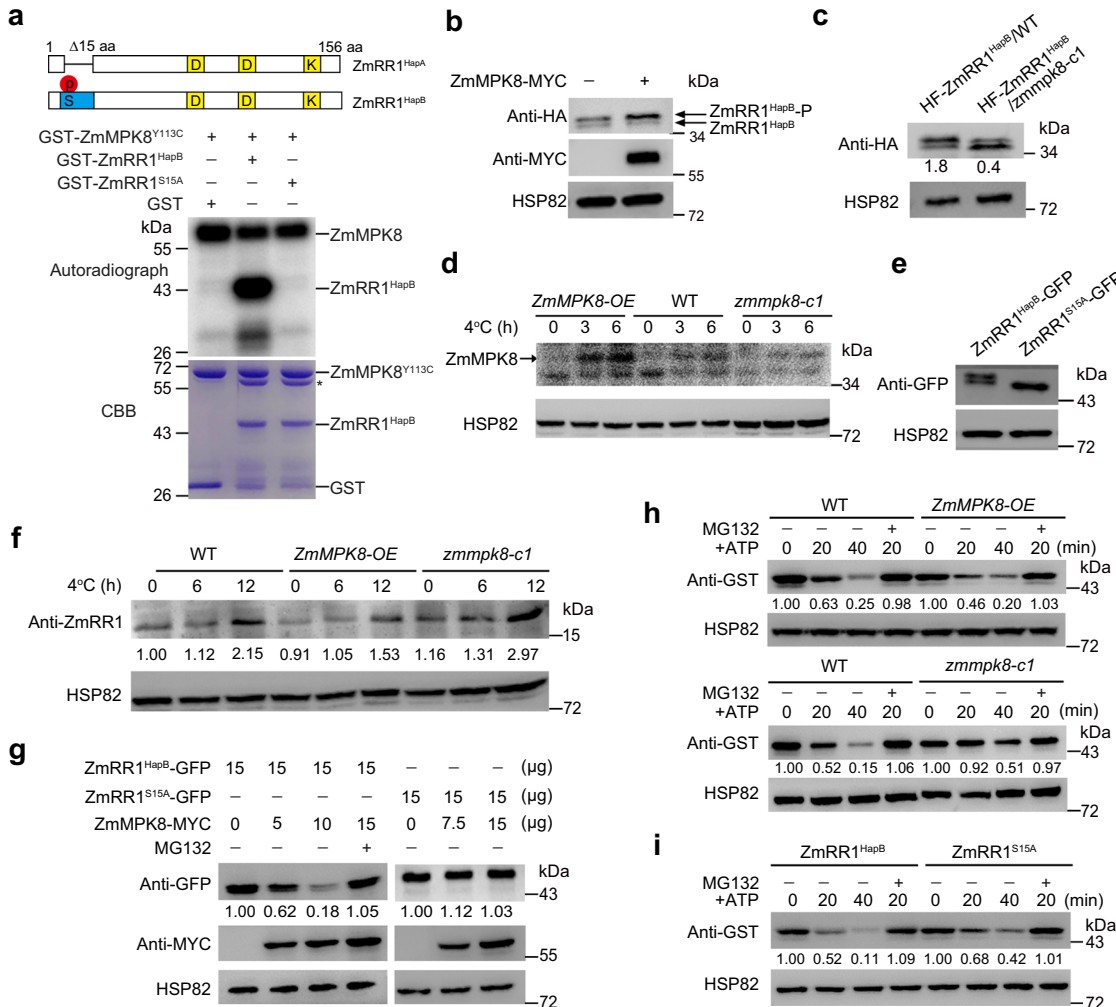

**Fig. 4 ZmMPK8 mediates the phosphorylation and degradation of ZmRR1. a** ZmMPK8 phosphorylates ZmRR1^HapB in vitro. Diagram shows the phosphorylation site on ZmRR1^HapB (top). Phosphorylated ZmRR1 was detected by autoradiography (middle). Recombinant proteins were detected by Coomassie brilliant blue (CBB) staining (bottom). Asterisk represents nonspecific bands. **b** Immunoblot analysis of HF-ZmRR1^HapB co-expressed with or without ZmMPK8-MYC in *N. benthamiana* leaves. **c** Immunoblot analysis of HF-ZmRR1^HapB expressed in maize protoplasts isolated from wild type or *zmmpk8-c1*. The relative ratio between phosphorylated and unphosphorylated protein was determined using ImageJ software. **d** In-gel kinase assay of ZmMPK8 using total protein extracts from the wild type, *ZmMPK8-OE*, and *zmmpk8* plants using GST-ZmRR1 as substrate. ZmMPK8 kinase activity was detected by autoradiography. HSP82 was used as a control. **e** ZmRR1^HapB-GFP and ZmRR1^S15A-GFP proteins expressed in maize protoplasts. ZmRR1 was detected with anti-GFP antibody. **f** Immunoblot analysis of ZmRR1^HapB protein levels in the wild type, *ZmMPK8-OE*, and *zmmpk8-c1* plants under cold stress. Ten-day-old seedlings were incubated at 4 °C for the indicated time. ZmRR1 was detected with anti-ZmRR1 antibody. **g** Immunoblot analysis of ZmRR1^HapB-GFP or ZmRR1^S15A-GFP co-expressed with increasing amounts of ZmMPK8-MYC in maize protoplasts. **h** Analysis of the stability of ZmRR1 protein in cell-free assays. GST-ZmRR1^HapB was incubated with equal amounts of total proteins extracted from 10-day-old wild type, *ZmMPK8-OE*, or *zmmpk8* seedlings in the presence of 10 mM ATP. GST- ZmRR1^HapB was detected with anti-GST antibody. **i** Analysis of the stability of ZmRR1^HapB and ZmRR1^S15A in a cell-free assay. In **g**–**i**, 50 μM of the 26 S proteasome inhibitor MG132 was used to determine ubiquitin-mediated protein degradation. In **a**–**i**, a representative experiment from three independent experiments is shown. Source data are provided as a Source Data file.

when co-expressed with ZmMPK8 (Fig. 4g). These data suggest that ZmMPK8-mediated phosphorylation of ZmRR1^HapB at Ser15 facilitates the degradation of ZmRR1^HapB via the 26 S proteasome pathway.

The degradation of ZmRR1^HapB was then examined in *ZmMPK8-OE* and *zmmpk8-c1* plants using a cell-free protein degradation assay. We expressed and purified recombinant GST-ZmRR1^HapB from *E. coli* and incubated it with ATP and equal amounts of total proteins extracted from wild type, *ZmMPK8*-overexpression, and *zmmpk8-c1* maize seedlings. Following incubation with ATP for the indicated time, GST-ZmRR1^HapB was gradually degraded in the presence of wild-type proteins (Fig. 4h). The degradation of GST-ZmRR1^HapB occurred much

more rapidly in the presence of proteins from *ZmMPK8-OE* plants, but more slowly in the presence of proteins from *zmmpk8-c1* mutant (Fig. 4h). Treatment with MG132 inhibited the degradation of ZmRR1^HapB (Fig. 4h), further supporting the notion that ZmRR1^HapB is subjected to 26 S proteasome-mediated degradation. To examine whether the phosphorylation of ZmRR1^HapB at Ser15 affects its stability, we incubated GST-ZmRR1^HapB or GST-ZmRR1^S15A with total proteins from *ZmMPK8-OE* plants. GST-ZmRR1^S15A degraded much more slowly than GST-ZmRR1^HapB (Fig. 4i). Taken together, these data suggest that ZmMPK8 phosphorylates ZmRR1^HapB to promote its ubiquitin-mediated degradation in maize under cold stress.

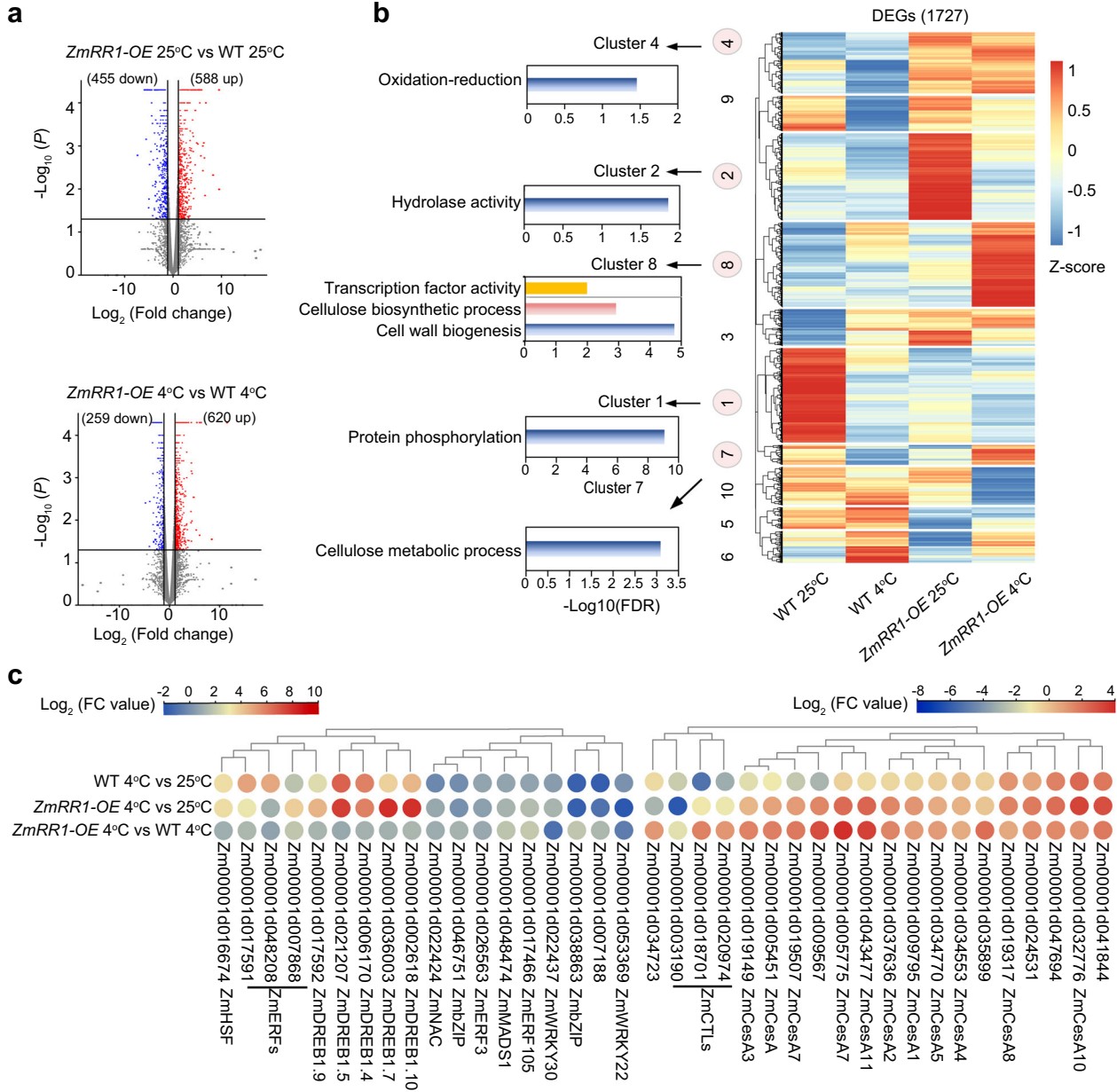

**Fig. 5 Transcriptome analysis of *ZmRR1*-regulated differentially expressed genes. a** Volcano plots showing the number of differentially expressed genes (DEGs) regulated by ZmRR1 at 25 and 4 °C. DEGs were identified using *P* value < 0.05 and absolute log2-fold change > 1 as criteria. **b** Hierarchical clustering and heatmap of 1727 ZmRR1-regulated DEGs and Key GO Terms. DEGs show segregation into 10 co-expressed clusters. The z-score scale represents mean-subtracted regularized log-transformed FPKM. Full results of GO enrichment analysis of heatmap DE clusters are shown in Supplementary Data 2. **c** Heatmap showing the DE transcription factors and cell-wall biogenesis related genes regulated by ZmRR1. Colors represent log2-fold change comparing relative expression.

**Transcriptome analysis of *ZmRR1*-regulated *COR* genes by RNA-sequencing.** To investigate the role of ZmRR1 in regulating chilling tolerance, we performed RNA-Seq analysis of 10-day-old wild-type LH244 and *ZmRR1-OE* maize seedlings incubated at 4 °C for 0 h or 12 h and identified differentially expressed genes in two independent experiments. Using RNA-seq data, we analyzed differentially expressed genes (i.e., ZmRR1-regulated genes) in the wild type vs. *ZmRR1-OE* plants based on the criteria fold change > 2 and *P* < 0.05 in *ZmRR1-OE* compared to the wild type under permissive conditions (25 °C) or under cold treatment (4 °C). At 25 °C, we identified 588 upregulated genes (ZmRR1-activated genes) and 455 downregulated genes (ZmRR1-repressed genes) in *ZmRR1-OE* compared to the wild type (Fig. 5a; Supplementary Data 2). After cold treatment, we identified 620 ZmRR1-activated

genes and 259 ZmRR1-repressed genes (Fig. 5a; Supplementary Data 3). A total of 1727 genes were identified as ZmRR1-regulated genes (Fig. 5b). Gene Ontology (GO) term analysis revealed these ZmRR1-regulated genes significantly enriched into 10 gene clusters (http://bioinfo.cau.edu.cn/agriGO/) (Supplementary Data 4). Functional annotations of individual clusters were associated with protein phosphorylation (cluster 1), hydrolase activity (cluster 2), oxidation reduction (cluster 4), cellulose and cell-wall metabolic process (clusters 7 and 8) (Fig. 5b).

Based on the defined cold-responsive transcriptome[43], up to 803 genes out of 1727 *ZmRR1*-regualted genes are significantly altered by cold treatment, which are termed as *ZmRR1*-regualted *COR* genes (Fig. 5c; Supplementary Fig. 10a and Supplementary

Data 5). In particularly, a set of *ZmDREB1* family genes encoding key transcriptional activators of *COR* genes have been identified (Fig. 5c; Supplementary Fig. 10b), suggesting that *ZmRR1* regulates maize chilling tolerance at least partially via *ZmDREB1*-dependent signaling pathway. Among the identified ZmRR1-regulated genes, the strongest enrichment was observed for genes associated with cell-wall biosynthesis genes (cluster 7 and cluster 8) (Fig. 5b). Intriguingly, the expression of these genes was dramatically increased in *ZmRR1-OE* seedlings, especially upon cold treatment (Fig. 5c). A detailed functional analysis of these genes revealed a strong enrichment for a set of *Chitinase-like protein* (*CTL*) and *Cellulose synthase A* (*CesA*) family genes (Fig. 5c). Thus, *ZmDREB1* and cell-wall-related genes may contribute to the enhanced chilling tolerance of *ZmRR1*-over-expression transgenic maize.

**ZmRR1 regulates chilling tolerance by modulating the expression of *ZmDREB1s* and *ZmCesAs*.** To confirm the expression changes of *ZmDREB1* and *ZmCesA* family genes between wild-type and *ZmRR1-OE* plants, quantitative real-time PCR analyses were performed under time-course cold treatment (Supplementary Fig. 11a). The results indicated that a significantly higher cold-induced upregulation of *ZmDREB1.7*, *ZmDREB1.10*, and *ZmCesA2*, *ZmCesA10* and *ZmCesA11* in *ZmRR1-OE* plants vs. the wild type, respectively (Fig. 6a, Supplementary Fig. 11b). By contrast, the cold-induced upregulation of *ZmDREB1s* and *ZmCesAs* were significantly suppressed in *zmrr1* vs. wild-type plants (Fig. 6b, Supplementary Fig. 11c). Given that ZmMPK8 negatively regulates the stability of ZmRR1, we examined the expression of *ZmDREB1s* and *ZmCesAs* in *ZmMPK8-OE* and *zmmpk8*. The cold-induced expression of *ZmDREB1s* and *ZmCesAs* was significantly lower in *ZmMPK8-OE*, but significantly higher in *zmmpk8* than the wild type (Fig. 6c, d; Supplementary Fig. 11d, e). These results indicate that the expression of *ZmDREB1s* and *ZmCesAs* was positively regulated by ZmRR1, but negatively regulated by ZmMPK8 in maize under cold stress.

To investigate whether the ZmMPK8- and ZmRR1-regulated expression of *ZmDREB1* and *ZmCesA* genes is responsible for chilling tolerance in maize, we generated transgenic lines overexpressing *ZmDREB1.10* and *ZmCesA2* and subjected two independent lines to chilling tolerance assays. As expected, *ZmDREB1.10-OE* lines displayed decreased relative injured area and ion leakage compared with the wild type following chilling stress (Fig. 6e–h). These results suggest that function of *DREB1* in enhancing plant cold tolerance is highly conserved in maize. *ZmCesAs* encode the catalytic subunits of cellulose synthase[44]. We observed that both *ZmCesA2-OE* independent transgenic lines displayed chilling-tolerant phenotypes, with decreased relative injury area and ion leakage compared to the wild type after cold treatment (Fig. 6i–l). These data demonstrate that *ZmCesA2* plays a positive role in chilling stress tolerance in maize.

## Discussion
In this study, we demonstrated that the MAPK-dependent phosphorylation of ZmRR1 confers chilling tolerance in maize. We utilized natural variation in a maize panel to identify an MPK phosphorylation residue on ZmRR1 associated with chilling tolerance in seedlings. There are three major findings of this study. First, the identification of ZmRR1 as a key component of the chilling tolerance mechanism sheds light on how maize integrates growth and cold stress adaptation. Second, the important role of the ZmMPK8 kinase in negatively regulating maize chilling tolerance has been revealed. Last but not the least, the variation of

ZmRR1^HapA is an optimal allelic variation that could be utilized for molecular breeding to improve cold tolerance in maize.

Cytokinin signaling involves a His-Asp phosphorelay, which is the key mechanism for plant growth and stress response[45]. The plant type-A RRs have been suggested to play important roles in mediating plant stress tolerance[46]. In *Arabidopsis* and rice, the expression of type-A *RRs* is rapidly induced by cytokinin as well as various abiotic stresses[34–36,47]. Consistently, the constitutive expression of type-A *RRs* not only negatively regulates cytokinin signaling but also confers plant abiotic stress tolerance. ZmRR1 encoding a maize type-A response regulator, has been shown to be induced by cytokinin and be necessary for negative feedback regulation of cytokinin signaling[48,49]. In this study, we demonstrated that ZmRR1 plays a crucial role in enhancing maize cold tolerance. Notably the accumulation of ZmRR1 protein but not the transcription is induced by cold stress. These observations suggest that cold may regulate ZmRR1 at post-translational level.

Association analysis indicated that the variation at the N-terminus of ZmRR1 (harboring a key phosphorylation residue) is correlated with chilling tolerance. Further analysis revealed that ZmRR1^HapA harbors a variation conferring increased chilling tolerance compared to ZmRR1^HapB due to the high stability of its nonphosphorylated form. Moreover, few defects in kernel phenotypes were observed in *ZmRR1*-overexpression lines under optimal feeding conditions (Supplementary Fig. 12), suggesting that ZmRR1^HapA would be an excellent allelic variation for use in maize breeding to improve cold tolerance. It has been shown that ZmRR1 contains a highly conserved D-D-K residue in its C-terminal receiver domain[41]. Notably, we found overexpressing *ZmRR1* led to moderately reduced plant height in field test (Supplementary Fig. 12a, b), which might be due to the repression of cytokinin signaling activity mediated by ZmRR1. This could improve lodging-resistance in high-yielding maize. Thus, we propose that ZmRR1 regulates cytokinin signaling via conserved C-terminus of ZmRR1, whereas the variable N-terminus of ZmRR1 likely employs a fine-tuned signaling pathway that transduces environmental signals to confer the divergent levels of stress resistance in maize inbred lines.

MAPK cascades are conserved signaling pathways present in all eukaryotes[40]. In the current study, we demonstrate that ZmMPK8, together with its paralog ZmMPK2[50], negatively regulates chilling tolerance. The MAPK pathway mediated by clade A MPK3/MPK6, clade B MPK4, and the upstream proteins MKK2/MKK4 are involved in plant responses to cold stress in *Arabidopsis* and rice[23,24,26,51,52]. ZmMPK2 and ZmMPK8 are only two of clade C MAPKs in maize, which lack the CD domain (a binding site for MAPKKs, protein substrates, and phosphatases) in their C-terminal regions, suggesting that ZmMPK2/8 might function downstream of the atypical MAPK kinase MKK[50]. Based on the observation that ZmMPK8 is activated by cold stress, the identification of critical MKK(s) in response to cold stress for the activation of ZmMPK8 is needed for further investigation. It appears that the cold-activated ZmMKK-ZmMPK2/8 cascade might employ a feedback attenuation mechanism to balance the contradictory effect of overwhelming cold-responsive gene expression to limit energy consumption under unfavorable conditions.

Whole-genome transcriptome analysis indicated that ZmRR1-mediated stress tolerance may involve in pathways like oxidation reduction, cell-wall metabolic process, and transcription factors (Fig. 5). More importantly, *ZmDREB1s*, encoding well-characterized core transcription factors of cold signaling, are upregulated in *ZmRR1-OE* plants. To elucidate whether ZmRR1 regulates the expression of *ZmDREB1* genes via the two-component system, we generated type-B *RR11*-overexpresing plants (Supplementary Fig. 13a) and examined their chilling

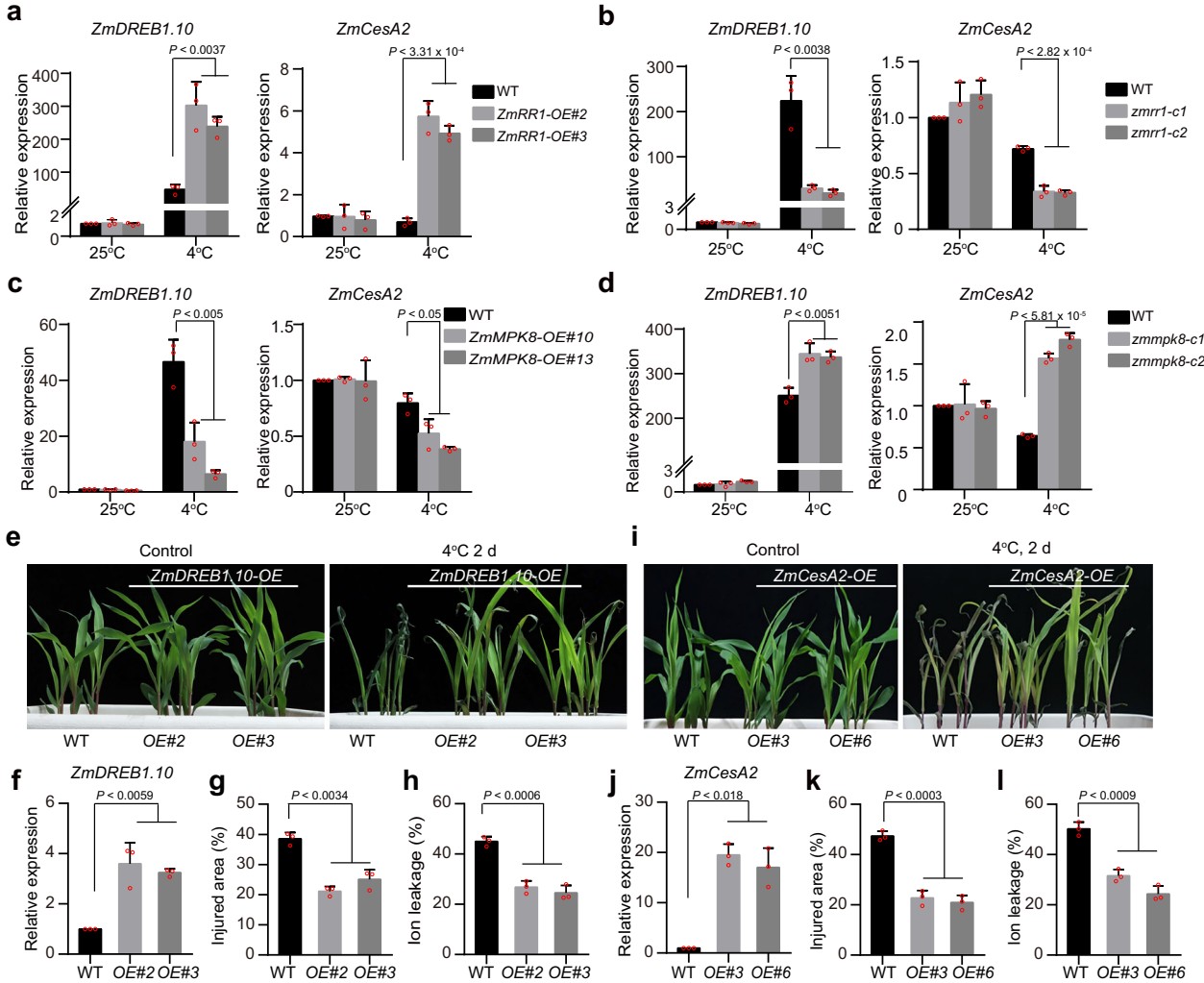

**Fig. 6 ZmRR1 regulates chilling tolerance by modulating the expression of *ZmDREB1* and *ZmCesA* genes. a–d** Relative expression levels of *ZmDREB1.10* and *ZmCesA2* in 10-day-old wild type, *ZmRR1-OE* (**a**) *zmrr1* mutant (**b**), *ZmMPK8-OE* (**c**), and *zmmpk8* plants (**d**) under cold treatment at 4 °C for 12 h. **e–h** Chilling phenotypes (**e**), *ZmDREB1.10* expression (**f**), injured area (**g**), and ion leakage (**h**) of *ZmDREB1.10* overexpression transgenic plants. **i–l** Chilling phenotypes (**i**), *ZmCesA2* expression (**j**), injured area (**k**), and ion leakage (**l**) of *ZmCesA2* overexpression transgenic plants. In **f** and **j**, *ZmDREB1.10* and *ZmCesA2* expression were examined in 25 °C-grown seedlings. In **a**, **b**, **c**, **d**, **f**, **j**, data are the mean values ± SD ($n = 3$; two-sided *t*-test). Three independent experiments were performed with similar results. In **g**, **h**, **k**, **l**, each bar represents the mean ± SD of three independent experiments. The statistical significance was determined by a two-sided *t*-test. Source data underlying Fig. 6a–d, f–h, and j-l are provided as a Source Data file.

phenotypes and *ZmDREB1* gene expression. Overexpression of *ZmRR11-OE* impaired maize chilling tolerance (Supplementary Fig. 13b, c); however, the cold-induced expression of *ZmDREB1s* in these transgenic plants was either upregulated or not altered (Supplementary Fig. 13d), which cannot account for the chilling sensitive phenotypes of *ZmRR11-OE* plants. Considering the previous findings that expression of *DREB1s* was not affected by cytokinin signaling under cold stress in *Arabidopsis*[34], it is possible that ZmRR1 regulates *DREB1* gene expression in a type-B independent manner. In addition, we also observed that ZmRR1 regulates expression of cell-wall remodeling genes, such as *ZmCesAs* and *ZmCTLs*. Similarly, *Arabidopsis* type-A ARR6 acts as a linkage for cell-wall composition and immune response[53]. Thus, type-A RRs plays important roles in controlling plant cell-wall composition under both biotic and abiotic stresses. As ZmRR1 is not transcription factors, it would be interesting to identify ZmRR1-associated transcription factors that directly regulate these *ZmCesA* and *ZmCTL* genes.

In summary, we propose a working model of the role of ZmRR1 in the response of maize to chilling stress (Fig. 7). Under

cold stress, ZmRR1 accumulates and induces the expression of *ZmDREB1s* and *ZmCesAs*, thus enhancing chilling tolerance. ZmRR1 is phosphorylated by cold-activated ZmMPK8 at Ser15, which promotes the ubiquitin-mediated degradation of ZmRR1. The stability of ZmRR1 protein is improved in an allele of ZmRR1[HapA] from natural maize varieties under cold stress due to the deletion of the ZmMPK8 phosphorylation site, thus providing a new strategy for breeding chilling-tolerant crops.

## Methods

**Plant materials**. The maize (*Zea mays* L.) transgenic plants were obtained from the Center for Crop Functional Genomics and Molecular Breeding, CAU. The construct was transformed with inbred line LH244. For association analysis, we used maize natural variation panel, which consists of inbred lines from tropical and temperate backgrounds[54,55].

**Phenotyping and statistical analyses of maize chilling tolerance at seedling stage**. Maize plants were grown in a greenhouse at 25 °C with a 16 h light/8 h dark photoperiod with ~200 μmol m$^{-2}$ s$^{-1}$ photon density and 60% relative humidity. Seeds were sown in a soil box (35 × 25 × 15 cm), and seedlings at V2 stage were exposed to a chilling stress treatment after watering thoroughly. The seedlings were treated at 4 °C for 2–4 days and subsequently recovered at 25 °C for additional 2 d.

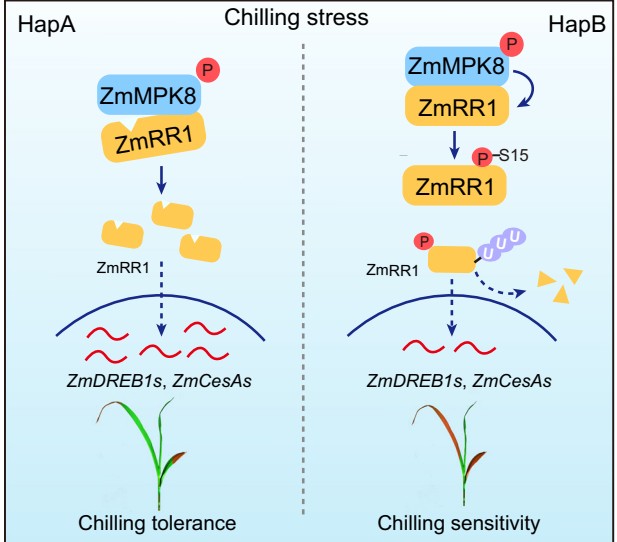

**Fig. 7 Proposed model of ZmRR1-mediated chilling tolerance.** Under cold stress, ZmRR1 protein accumulates to enhance chilling tolerance. ZmMAPK8 interacts with and phosphorylates ZmRR1^HapB at Ser15 to promote its ubiquitin-mediated degradation under chilling stress. The 45 bp deletion of ZmRR1^HapA prevents its degradation by ZmMPK8, leading to the higher abundance of ZmRR1^HapA. As a result, HapA maize inbred lines have increased expression of *ZmDREB1* and *ZmCesA* genes, and thus enhanced chilling tolerance vs. HapB.

Then the second leaves were photographed and measured by ImageJ software, respectively. The relative injury area was calculated with following formula: Relative injured area (%) = A2/A1 × 100; A1 = full area of the second leaf; A2 = injured area of the second leaf. *Arabidopsis thaliana* Col-0 ecotype was used in freezing tolerance assay. 14-day-old *Arabidopsis* seedlings grown on half-strength MS plates at 22 °C containing 0.8% agar and 2% sucrose under long day conditions (16 h light/8 h dark, 100 µE m$^{-2}$ s$^{-1}$) were treated at −5 °C for 1 h (non-cold acclimation, NA) or were treated at −10 °C for 1 h after pretreated at 4 °C for 3 days (cold acclimation, CA)[19]. Then the seedlings were transferred to 4 °C in the dark for 12 h and recovered at 22 °C for 3 days before calculating the survival rates of seedlings.

For ion leakage assays, the leaves of chilling-treated maize seedlings were collected into 15 mL tubes containing 10 mL deionized water. The solution was vacuumed for 30 min, and the conductance of the water was measured as S0. After shaken at room temperature for 1 h, the solution was detected as S1. Then the samples were boiled for 30 min and shaken at room temperature for cooling, followed by detecting S2. The value of (S1 − S0)/(S2 − S0) was calculated as ion leakage.

**ZmRR1-based association analysis.** A total of 558,629 SNPs with a minor allele frequency (MAF) >0.05 were generated by Illumina MaizeSNP50 array covering the whole maize genome[54,56]. The standard mixed linear model was applied (TASSEL 3.0)[57], in which the population structure (Q) and kinship (K) were estimated[56] for ZmRR1-based association analysis. P value was calculated by the mixed linear model (MLM, Tassel3.0). One hundred sixty one maize inbred lines were analyzed for the association between the genetic variations in *ZmRR1* and the injured area. The *ZmRR1* coding regions and 5′UTR sequences (545 bp containing 74 bp 5′-UTR and 471 bp coding region) were amplified and sequenced. The sequences were assembled by MEGA 7.0 software. DNA sequence polymorphisms (SNPs and InDels) were identified, and then the association between variations (exclude synonymous mutations) and the leaf injured area, as well as the pairwise LD were calculated by TASSEL 3.0[58,59]. Two independent replicates were performed to measure the relative leaf injured ratio, and the mean value was conducted for association mapping.

**Plasmid construction and plant transformation.** To generate *Super:ZmRR1-GFP*, *Super:ZmRR1-MYC*, *Super:ZmMPK8-GFP*, and *Super:ZmMPK8-MYC* constructs, the coding sequences (CDSs) of *ZmRR1* and *ZmMPK8* were amplified and cloned into Super1300-GFP (KpnI) using in-fusion PCR cloning systems (Clone smarter). To generate the *35S:HA-Flag-ZmRR1* construct, the *ZmRR1* CDS was amplified and cloned into the pCM1307 vector.

To generate *GST-ZmRR1*, *GST-ZmMPK8*, and *His-ZmMPK8* constructs, the CDSs were cloned into pGEX4T-1 (EcoRI) and pET32a (BamHI) vectors using in-fusion PCR cloning systems. Using *ZmMPK8-pGEX-4T-1* and *ZmRR1-pGEX4T-1*

plasmids as templates, site-directed mutagenesis was carried out to obtain *GST-ZmMPK8^Y113C* and *GST-ZmRR1^S15A* constructs.

To generate *ZmRR1-BD*, *ZmMPK8-N (1–170 aa)-BD*, *ZmMPK8-KD (171–369 aa)-BD*, and *ZmMPK8-AD*, the full-length or truncated CDS of *ZmRR1* and of *ZmMPK8* were amplified and cloned into pGBKT7 (EcoRI) or pGADT7 (EcoRI) by using in-fusion PCR cloning systems.

The cDNA of *ZmRR1* was cloned into the pUC-SPYCE (BamHI) vector to generate *ZmRR1-YFP^C*. Full-length *ZmMPK8* and truncated *ZmMPK8* were cloned into the pUC-SPYNE (BamHI) to generate *ZmMPK8-YFP^N* and *ZmMPK8-YFP^N (1–170 aa)*.

To generate maize transgenic plants, the CDS of *ZmRR1*, *ZmMPK8*, *ZmDREB1.10*, *ZmCesA2*, and *ZmRR11* were amplified. By incubating with Taq polymerase, the 3′A overhangs were added to the PCR products, and then ligated into the linearized pBCXUN vector (XcmI) containing the complementary 3′-T using Solution I ligase (Takara).

To generate CRISPR/Cas9 knockout lines of *ZmRR1* and *ZmMPK8*, the target sites located at the first exons of *ZmRR1* and *ZmMPK8* were obtained from CRISPR-P (http://crispr.hzau.edu.cn/CRISPR2/). Oligo-F and Oligo-R were slowly annealed and inserted between the two BsaI sites of pBUE411[60]. The constructs were transformed into the *Agrobacterium tumefaciens* strain EAH105. The immature zygotic embryos (12 day after pollination) of the maize inbred line LH244 were used for *Agrobacterium*-mediated maize transformation to generate transgenic lines[61]. All primes used were list in Supplementary Data 6.

*pSuper:ZmRR1^HapA-GFP* and *pSuper:ZmRR1^HapB-GFP* constructs were transformed into wild-type *Arabidopsis thaliana* plants by floral dip. Transgenic plants were selected by hygromycin B and T3 homozygous transgenic plants were used in this study.

**qRT-PCR and RNA-seq assays.** Total RNAs were extracted from 10-day-old seedlings grown on soil with TRIzol reagent (Invitrogen), followed by reverse transcription with M-MLV reverse transcriptase (Promega). Then quantitative real-time PCR assays were performed with SYBR Green regent (Takara) on a 7500 Real-Time PCR system (Applied Biosystem) to detect specific genes. The expression of *UBI* was used as a standard control[62].

For RNA-seq analysis, 10-day-old maize seedlings (V2 stage) with or without 12 h of 4 °C treatment were collected. Maize plants were grown in a greenhouse at 25 °C with a 16 h light/8 h dark photoperiod with ~200 µmol m$^{-2}$ s$^{-1}$ photon density and 60% relative humidity. Seeds were sown in a soil box (35 × 25 × 15 cm), and seedlings at V2 stage were quickly move to 4 °C light incubator for a 12 h cold treatment. The conditions such as photoperiod, photon density and relative humidity in the incubator were the same as in the greenhouse. The RNA materials of 25 °C and 4 °C were obtained by the following methods. The second leaf of the three seedlings were cut into the mixing pool and then frozen in liquid nitrogen, followed by RNA extraction. Two independent replicates were performed.

Total RNA was isolated using TRIzol reagent (Biotopped) and RNA integrity was evaluated using a Bioanalyzer 2100 (Agilent). The libraries were sequenced on an illumina nova 6000 at Berry Genomics (Beijing) and 150 bp paired-end reads were generated. Raw data (raw reads) of fastq format were firstly subjected to quality control using FastQC (v.0.11.9). Then reads were mapped to the Maize genome (B73 RefGen_v4, AGPv4) using HISAT2 (v.2.2.0) with default parameters. FPKM of each gene was calculated using Cufflinks (v.2.2.1). The differential-expression analysis was performed using Cuffdiff (v.2.2.1) with default parameters. Significant DEGs were identified as those with a P value (one-way ANOVA test) of differential expression above the threshold (log2 > 1, P < 0.05). GO enrichment was performed with the accession numbers of significant DEGs via agriGO v2.0. RNA-seq dataset has been deposited in NCBI accessible database.

**ZmRR1-antibody production and immunoblot analysis.** Peptide containing 15 amino acids (SSPKAAGDNRKTVVS) at the N-terminus of ZmRR1 was selected to immunize New Zealand white rabbits in Shanghai Youke Biotechnology. In the immunoblot assays, the antibody dilution ratio is 1:1000.

Total proteins were extracted from 10-day-old seedlings or protoplasts with protein extraction buffer (50 mM Tris-HCl, pH 7.5, 150 mM NaCl, 10 mM MgCl₂, 0.2% NP-40, 5 mM DTT and 1×protease inhibitor cocktail). 5×SDS loading buffer was added into the protein samples, and the mixtures were incubated at 100 °C for 5 min. After separation on SDS-PAGE, ZmRR1 protein was detected with anti-ZmRR1, anti-GFP (Abmart, Cat#M20004, 1:5000), anti-HA antibodies (Sigma–Aldrich; Cat#H3663, 1:5000). anti-MYC antibodies (Sigma–Aldrich; Cat#M4439, 1:5000).

**Subcellular localization.** To detect the subcellular localization of ZmRR1 and ZmMPK8 proteins, *Super:ZmRR1-GFP*, *Super:ZmMPK8-GFP*, and *pSuper1300-GFP* vectors were transformed into maize protoplasts and incubated at 22 °C for 15 h. GFP fluorescence was observed by confocal microscopy (ZEISS710; Carl Zeiss) with exciting light 488.

**Yeast two-hybrid assay.** Constructs were co-transformed into AH109 yeast cells. The empty vector *pGADT7* and *pGBKT7* were co-transformed as negative controls.

The interaction was determined on synthetically defined (SD)/-Ade/-His/-Leu/-Trp medium following the manufacturer's protocols (Clontech).

**Co-immunoprecipitation assay**. *ZmMPK8-MYC* and *ZmRR1-GFP* vectors were co-transformed into wild-type maize protoplasts, and kept in darkness for 15 h. Total proteins were extracted with IP buffer (50 mM Tris-HCl, pH 7.5, 150 mM NaCl, 0.2% NP-40, 5 mM DTT, and 1×protease inhibitor cocktail) and incubated with GFP beads (Chromo Tek) for 2 h. The immunoprecipitated samples were washed five times with washing buffer (50 mM Tris-HCl, pH 7.5, 150 mM NaCl, 0.1% NP-40), separated on SDS-PAGE and subjected to immunoblot analysis with anti-MYC antibody (Sigma–Aldrich).

**BIFC assay**. Vectors of *ZmMPK8-YFP$^N$*, *ZmMPK8(1–170 aa)-YFP$^N$* and *ZmRR1-YFP$^C$* were co-transformed into wild-type maize protoplasts and incubated at 22 °C for 15 h. GFP fluorescence was observed by confocal microscopy (ZEISS710; Carl Zeiss) with exciting light 488.

**In vitro pull-down assay**. Recombinant proteins GST-ZmRR1 and His-ZmMPK8 were purified from *E. coli*. GST-ZmRR1 or GST were incubated with Glutathione beads (GE Healthcare) at 4 °C for 2 h, and subsequently incubated with His-ZmMPK8 for 1 h. After eluted from the beads, the proteins were subjected to immunoblot analysis with anti-His antibody (Beijing Protein Innovation, Cat#-AbM59012 18 PU, 1:5000) to detect His-ZmMPK8.

**In vitro phosphorylation assay**. In vitro phosphorylation assay was performed as reported previously[19]. Briefly, 1 µg recombinant proteins His-ZmMPK8 and 10 µg GST-ZmRR1 were incubated in kinase reaction buffer (20 mM Tris-HCl pH 7.5, 20 mM MgCl$_2$, 1 mM DTT, 50 mM ATP) at 30 °C for 30 min with or without 1µCi [γ-$^{32}$P] ATP. The proteins were separated by SDS-PAGE and visualized by autoradiography.

**Cell-free degradation assay**. Cell-free degradation assay was performed as reported previously[63]. Briefly, total proteins were extracted from 10-day-old seedlings with native buffer (50 mM Tris-MES pH 8.0, 10 mM EDTA pH 8.0, 0.5 M Sucrose, 1 mM MgCl$_2$, 5 mM DTT). Purified recombinant GST-ZmRR1 was added into equal amount of WT, *zmmpk8-c1* and *ZmMPK8-OE* total proteins with 10 mM ATP and incubated at 25 °C for different time periods, and 50 µM MG132 was used. GST-ZmRR1 was separated on SDS-PAGE and detected with anti-GST antibody (Beijing Protein Innovation, Cat#AbM59001 2H5 PU, 1:5000).

**In-gel kinase assays**. In-gel kinase assays were performed as reported previously[19]. Briefly, total proteins were extracted from 10-day-old seedlings (control or cold-treated at 4 °C) with protein extraction buffer containing 5 mM EDTA pH 8.0, 5 mM EGTA pH 8.0, 25 mM NaF, 1 mM Na$_3$VO$_4$, 20% (v/v) glycerol, 2 mM DTT, 1×protease inhibitor cocktail (Roche) and 25 mM HEPES-KOH pH 7.5. The proteins were separated on a 10% (v/v) SDS-PAGE gel containing 0.5 mg/ml GST-ZmRR1. Then, the gel was washed three times with washing buffer (25 mM Tris-HCl pH 7.5, 0.5 mM DTT, 5 mM NaF, 0.1 mM Na$_3$VO$_4$, 0.5 mg/ml BSA and 0.1% [v/v] Triton X-100) at room temperature for 20 min each. The gel was incubated in renatured buffer containing 25 mM Tris-HCl pH 7.5, 1 mM DTT, 5 mM NaF, and 0.1 mM Na$_3$VO$_4$ at 4 °C for 1 h, 12 h, and 1 h, respectively. Next, the gel was incubated in kinase reaction buffer (40 mM HEPES-KOH pH 7.5, 1 mM DTT, 12 mM MgCl$_2$, 0.1 mM Na$_3$VO$_4$, and 2 mM EGTA) at room temperature for 30 min and then incubated in new kinase reaction buffer supplemented with 70 µCi [γ-$^{32}$P] ATP and 9 µl 1 mM cold ATP at room temperature for 2 h. The gel was washed by 5% (w/v) TCA and 1% (w/v) sodium pyrophosphate for five times (30 min for each). The signal was detected by Typhoon 9410 imager[64].

**Mass spectrometry assays**. To identify the phosphorylation site of ZmRR1 by ZmMPK8 in vitro, 10 µg GST-ZmRR1, and 1 µg His-ZmMPK8 purified proteins were incubated in 20 µl of protein kinase reaction buffer containing 20 mM MgCl$_2$, 50 mM Tris-HCl pH 7.5, 1 mM DTT, and 50 µM ATP, at 30 °C for 30 min. The reaction products were reduced by DTT and alkylated by IAM (Iodoacetamide), followed by digestion with trypsin (pH 8.5) at 37 °C for 12 h. The results were analyzed by LC-MS/MS[22].

For LC-MS/MS analysis in planta, total proteins were extracted from maize protoplasts expressing ZmRR1-MYC with protein extraction buffer (50 mM Tris-HCl, pH 7.5, 150 mM NaCl, 10 mM MgCl$_2$, 0.2% NP-40, 5 mM DTT, and 1×protease inhibitor cocktail) and incubated with MYC beads for 2 h. The immunoprecipitated samples were washed five times with protein extraction buffer and then eluted the proteins from beads and perform the LC-MS/MS assay.

## Data availability

Data supporting the findings of this work are available within the paper and its Supplementary Information files. A reporting summary for this Article is available as a Supplementary Information file. The RNA-seq data generated in this study have been

deposited in the National Center for Biotechnology Information Sequence Read Archive database under accession PRJNA344653. The mass spectrometric data generated in this study have been deposited in Integrated Proteome Resources with the project IPX0003279000. The raw sequencing data of maize under cold stress have been previously deposited at the NCBI under accession PRJNA344653[65]. Source data are provided with this paper.

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

## Acknowledgements

We thank Xiaohong Yang for providing inbred lines and helpful discussion, and Zhen Li for helping with the LC-MS analysis. The transgenic seeds of maize were created by Center for Crop Functional Genomics and Molecular Breeding of China Agricultural University. This work was supported by grants from State's Key Project of Research and Development Plan (2016YFD0100605), the National Key Research and Development Project (2020YFA0509902), and the National Natural Science Foundation of China (32022008 and 31921001).

## Author contributions

S.Y. conceived, designed, and directed the project. R.Z., Z.L. and Y.S. performed most of the experiments. P.Y. and J.C. performed the maize transformation. R.Z., Y.S., and D.F. performed association and RNA-seq analysis. R.Z., Z.L., C.J. and Y.S. analyzed the data. R.Z., Y.S. and S.Y. wrote the manuscript with comments from all authors.

## Competing interests

The authors declare no competing interests.
