## [Peer Review File · Nature Communications]

REVIEWER COMMENTS

Reviewer #1 (Remarks to the Author):

Here, Zeng and colleagues report that the type-A RR ZmRR1 is a positive regulator of maize chilling tolerance. Interestingly, they also show that mutations in ZmRR1 contribute to the natural variation of chilling tolerance in maize. Molecular characterization of one of these variants revealed that ZmRR1 accumulation is controlled by its phosphorylation. Through a two-hybrid screening, they have found that ZmRR1 interacts with ZmMPK8. This protein kinase negatively controls maize chilling tolerance by phosphorylating ZmRR1 to promote its degradation. Finally, Zeng and coworkers show that increased expression of ZmRR1 provoke a wide transcriptomic reprogramming. They found that a subset of ZmCesA gene family is up regulated in ZmRR1-OE plants. Characterization of maize plants overexpressing ZmCesA2 demonstrates that this gene is a positive regulator of chilling tolerance. They conclude that increased expression of ZmCesA genes would account for part of the increased cold tolerance of ZmRR1-OE plants.

Low temperature is one of the most relevant environmental stresses limiting crop productivity world wide. Obtaining new stress-tolerant crop varieties is essential to ensure food security in a climate change scenario. A mandatory step to succeed in this challenge is to understand the mechanisms evolved by plants to face environmental stress such as low temperatures and survive. Zeng and colleagues work not only provides very valuable data to unravel the regulation of these mechanisms, but it also demonstrates that ZmRR1 could be a very useful tool for molecular breeding in maize. The experimental approaches are sound and well performed. The results obtained completely support the conclusions. I consider that the data reported is of general interest. I only have few minor comments to the authors.

1. I think that authors should give more information in the introduction about previous knowledge of the function of type-A RR proteins in plants, and about their implication in the control of plant response to abiotic stress.
2. Authors state that ZmDREB1s proteins play a positive role in plant response to cold stress (lines 294). Although it is well known that DREB1 transcription factors are essential for the adaptation to low temperature in different plant species, I could not find any data about their role in maize response to cold. In fact, the references used to support this conclusion in Discussion are of works done with Arabidopsis. Authors should clarify whether or not there is any data about ZmDREB1s implication in maize response to cold.
3. Transactivation experiments are usually performed to assess whether a protein directly binds to the promoter of a target gene to control its expression. However, I do not see any result in the manuscript suggesting that this is the mechanisms followed by ZmRR1 to control ZmDREB1.7 expression. Authors need to justify why they performed this experiment. All in all, I think that removing this experiment would benefit to the clarity of the manuscript.
4. In line 351 authors suggest that it could be interesting to analyze whether ZmRR1 expression is controlled by b-type RRs. Although I do not have any doubt about the interest of this analysis, the limited description of the function of RRs regulators provided in the introduction hampers the evaluation of the relevance of this study.
5. The description of the model in figure 6 needs to be improved to facilitate its understanding. For

instance, there is not any description of what HK or HP stand for.

6. I miss a hypothesis about the molecular mechanisms followed by ZmRR1 to control cold-induced gene expression.

Reviewer #2 (Remarks to the Author):

The authors presented the role of the type-A Response Regulator (ZmRR1) in chilling stress tolerance in maize. The authors used transgenic and CRISPR/cas9-mediated mutation approaches along with biochemical and molecular biological methods, demonstrating that the type-A RR ZmRR1 is phosphorylated at Ser15 by ZmMPK8, which facilitates its ubiquitination-mediated degradation under cold stress, thereby negatively regulating chilling tolerance in maize. They further showed that a natural variation of ZmRR1 lacking a 15-amino acid region harboring Ser15 cannot be phosphorylated by ZmMPK8, resulting in enhanced ZmRR1 protein stability and chilling tolerance. They also found that ZmRR1 positively regulates ZmDREB1 and Cesa gene expression to enhance chilling tolerance. Finally the authors claimed that these findings could be a potential genetic resource for modifying chilling tolerance traits in crops.

The authors reported interesting results on how the type RR regulates chilling stress tolerance in maize and provided a potential mean of improving crop yield under chilling stress. However, how ZmRR1 upregulates ZmDREB1 and Cesa genes under chilling stress is unclear, given the fact that in Arabidopsis, the type-A ARRs act as negative feedback regulators of the type-B ARR transcription factors, thus inhibiting cytokinin and environmental stress-responsive gene expression to attenuate these responses. This question is critical to this manuscript, as it will provide molecular and genetic understanding of how ZmRR1 controls chilling stress in maize. As minor issues, discussion should be strengthened by comparing and contrasting the previous findings on two-component signaling system in Arabidopsis and other plants. Is ZmDREB1 known to protect chilling stress in maize?

Reviewer #3 (Remarks to the Author):

In this manuscript, Zeng and colleagues search for regulators of chilling tolerance in maize. They identify ZmMPK8 as a negative regulator of chilling tolerance that interacts with ZmRR1 and phosphorylates it. They connect ZmRR1 to the expression of chilling responsive genes. This is a well written paper and contains elegant experiments, but the final bioinformatic analysis of the response to cold is disappointing and should be improved.

(1) The authors speculate that the ZmMPK8 repression of ZmRR1 has evolved in order to avoid excess activation of the cold stress pathway. If this is the case, mutants in *zmpmk8* and the unphosphorylatable versions of ZmMPK8 which are more cold tolerant should have a yield penalty. Is this the case?

(2) The labelling of Fig. 4 is very poor. What does CK refer to? Presumably wild-type, but what does CK mean? It says "cold treatment" but doesn't say the temperature.

(3) Since the data is already there, it seems it would be quite easy to show whether or not there is a

systematic alteration of the cold responsive transcriptome. This isn't shown here. It is not clear for example if there is a control for the 12 h treatment. 0 h is not a good control because lots of circadian and diurnal genes will be differentially regulated even at 12 h of non-cold temperature. If it could be shown that the team are able to identify the cold responsive transcriptome (even if they have to use another paper to define these genes), then they should show how these 100 or so genes are differentially expressed in their conditions with these genotypes, that would be very helpful.

(4) In Fig. 4 it would make sense to actually have a comprehensive analysis of the RNA-seq data. It seems the reporter assays (e-j) would be better in a separate figure or in the supplemental material. As it is presented, the RNA-seq data is not clear. Similarly, the GO-term enrichments bar chart could also go to the supplementary material. Given that the RNA-seq has been performed the authors should show whether or not it indicates a meaningful mis-regulation of the cold regulon in the different genetic backgrounds.

(5) In Fig. 4b the legend states: heat map showing the log ratios of DEGs, but each column is for one experimental condition. There is no indication of a ratio. For example it says CK_0h. What is CK? How is this referring to a ratio? A ratio requires two values to be compared, and yet the descriptors refer to individual samples.

(6) In line 290 it says "Heatmap and cluster analyses indicated..." I cannot see any cluster analysis. This is an excellent idea, and I do think the clustering analysis should be performed and presented.

(7) Figure 4 d is presumably using a ratio, but the scale bar is very confusing. The colour scheme is very hard to follow, as is the range. It appears to show that ZmRR1-OE has almost no effect compared to WT on the cold response. This is why a much more comprehensive analysis as described in (3) above is necessary.

(8) The columns in Fig 5a are not labelled in the fig or legend.

(9) The same description is used for Fig 4b,4d and 5a ('log ratios') but the scale bar in Fig 4d is different from Fig 4b,5a. And column names for these 3 figures are totally different (4b, no 'vs'; 4d, 'vs'; 5a, nothing).

Responses to comments by Editor and Reviewers

Reviewer #1 (Remarks to the Author):

Low temperature is one of the most relevant environmental stresses limiting crop productivity world wide. Obtaining new stress-tolerant crop varieties is essential to ensure food security in a climate change scenario. A mandatory step to succeed in this challenge is to understand the mechanisms evolved by plants to face environmental stress such as low temperatures and survive. Zeng and colleagues work not only provides very valuable data to unravel the regulation of these mechanisms, but it also demonstrates that ZmRR1 could be a very useful tool for molecular breeding in maize. The experimental approaches are sound and well performed. The results obtained completely support the conclusions. I consider that the data reported is of general interest. I only have few minor comments to the authors.

1. I think that authors should give more information in the introduction about previous knowledge of the function of type-A RR proteins in plants, and about their implication in the control of plant response to abiotic stress.

Response: We thank the reviewer for the valuable comments and suggestions. We described the function of type-A RR in regulating plant stress response in the Introduction part as follows: *“Response regulators majorly consist of type-A and type-B RRs. Type-B RRs are transcription factors that can directly promote the expression of type-A RR genes. Type-A RRs, which contain a receiver domain but lack of output domain, are responsible for repressing cytokinin signaling via a negative feedback loop (Hwang et al., 2012). It has been shown that type-A RRs are extensively involved in abiotic stress responses, such as drought and cold stress responses. For example, SnRK2s phosphorylate the Ser residue of ARR5 to enhance its stability and plant drought tolerance (Huang et al., 2018). Cold induces the expression of ARR5,6,7,15 genes to enhance the freezing tolerance in Arabidopsis (Jeon et al., 2010; Shi et al., 2012). In rice, OsRR6 is induced by various abiotic stresses and enhances*

drought and salinity tolerance, while OsRR9 and OsRR10 negatively regulate salinity tolerance in rice (Wang et al., 2019a; Bhaskar et al., 2021). Nevertheless, the mechanism underlying type-A RRs regulate cold stress and type-A RRs is regulated under cold stress in crops remain unclear.”

2. Authors state that ZmDREB1s proteins play a positive role in plant response to cold stress (lines 294). Although it is well known that DREB1 transcription factors are essential for the adaptation to low temperature in different plant species, I could not find any data about their role in maize response to cold. In fact, the references used to support this conclusion in Discussion are of works done with Arabidopsis. Authors should clarify whether or not there is any data about ZmDREB1s implication in maize response to cold.

Response: We do agree with this reviewer that exploring the function of ZmDREB1s in regulating maize chilling tolerance is important for this study. In the revised version, we have added the chilling phenotype of *ZmDREB1.10* overexpression transgenic maize to verify that ZmDREB1.10 plays a positive role in plant tolerance to cold stress (Fig. 5). We have described this result as follows: “*As expected, ZmDREB1.10-OE lines displayed decreased relative injury area and ion leakage compared with the wild type following chilling stress (Fig. 6e-h). These results indicate that function of DREB1 in enhancing plant cold tolerance is highly conserved in maize.*”

3. Transactivation experiments are usually performed to assess whether a protein directly binds to the promoter of a target gene to control its expression. However, I do not see any result in the manuscript suggesting that this is the mechanisms followed by ZmRR1 to control ZmDREB1.7 expression. Authors need to justify why they performed this experiment. All in all, I think that removing this experiment would benefit to the clarity of the manuscript.

Response: We thank the reviewer for this good suggestion, and removed the data in our revised manuscript.

4. In line 351 authors suggest that it could be interesting to analyze whether ZmRR1 expression is controlled by b-type RRs. Although I do not have any doubt about the interest of this analysis, the limited description of the function of RRs regulators provided in the introduction hampers the evaluation of the relevance of this study.

Response: We thank the reviewer for pointing out this misleading sentence. In our revised manuscript, we added the description about cytokinin signaling including B-type RRs in Introduction, and deleted the description of irrelevant hypothesis in the Discussion.

5. The description of the model in figure 6 needs to be improved to facilitate its understanding. For instance, there is not any description of what HK or HP stand for.

Response: We thank the reviewer for pointing this out. We have modified the model in Figure 7 as suggested. To make it clearer, we deleted the model at warm temperature and emphasized the function of ZmRR1 in regulating chilling tolerance.

6. I miss a hypothesis about the molecular mechanisms followed by ZmRR1 to control cold-induced gene expression.

Response: We greatly appreciate this valuable suggestion. To address this concern, we generated type-B RR11-overexpression lines to dissect whether ZmRR1 regulates *COR* genes via the type-B ARR transcription factors. The results shown in Supplementary Fig. 13 suggest that it might be not the case. We have thoroughly revised the discussion part and described the potential mechanisms of ZmRR1 in regulating cold-induced gene expression as follows: *Whole-genome transcriptome analysis indicated that ZmRR1-mediated stress tolerance may involve in pathways like oxidation reduction, cell wall metabolic process, and transcription factors (Fig 5). More importantly, ZmDREB1s, encoding well-characterized core transcription factors of cold signaling, is up-regulated in ZmRR1-OE plants. To elucidate whether ZmRR1 regulates the expression of ZmDREB1 genes via the two-component system, we generated type-B RR11-overexpressing plants (Supplementary Fig. 13a) and*

examined their chilling phenotype and ZmDREB1 gene expression. Overexpression of ZmRR11-OE impaired maize chilling tolerance (Supplementary Fig. 13b,c). In contrast, the cold-induced expression of ZmDREB1 in these transgenic plants was either upregulated or not altered (Supplementary Fig. 13d), which cannot explain the chilling sensitive phenotype of ZmRR11-OE plants. Considering the previous finding that expression of DREB1s was not affected by cytokinin signaling under cold stress in Arabidopsis (Jeon et al., 2010), we suggest that ZmRR1 regulates DREB1 expression possibly in a type-B independent manner. In addition, we also observed that ZmRR1 regulates expression of cell-wall remodeling genes, such as ZmCesAs and ZmCTLs. Similarly, Arabidopsis type-A ARR6 acts as a linkage for cell-wall composition and immune response (Bacete et al., 2020). Thus, type-A RRs plays important roles in controlling plant cell wall composition under biotic and abiotic stresses. As ZmRR1 is not transcription factors, it would be interesting to identify ZmRR1-associated transcription factors that directly regulate these ZmCesA and ZmCTL genes.

Reviewer #2 (Remarks to the Author):

The authors reported interesting results on how the type RR regulates chilling stress tolerance in maize and provided a potential mean of improving crop yield under chilling stress. However, how ZmRR1 upregulates ZmDREB1 and CesA genes under chilling stress is unclear, given the fact that in Arabidopsis, the type-A ARRs act as negative feedback regulators of the type-B ARR transcription factors, thus inhibiting cytokinin and environmental stress-responsive gene expression to attenuate these responses. This question is critical to this manuscript, as it will provide molecular and genetic understanding of how ZmRR1 controls chilling stress in maize. As minor issues, discussion should be strengthened by comparing and contrasting the previous findings on two-component signaling system in Arabidopsis and other plants. Is ZmDREB1 known to protect chilling stress in maize?

Response: We thank the reviewer for evaluation of our manuscript and provide

valuable suggestions. In response to the concern of the reviewer, we have made following modification in Discussion and Result part:

(1) We generated type-B RR11-overexpression lines to dissect whether ZmRR1 regulates *COR* genes via the type-B ARR transcription factors. The results shown in Supplementary Fig. 13 suggest that it might be not the case. The detailed descriptions were as follows: *“Whole-genome transcriptome analysis indicated that ZmRR1-mediated stress tolerance may involve in pathways like oxidation reduction, cell wall metabolic process, and transcription factors (Fig 5). More importantly, ZmDREB1s, encoding well-characterized core transcription factors of cold signaling, is up-regulated in ZmRR1-OE plants. To elucidate whether ZmRR1 regulates the expression of ZmDREB1 genes via the two-component system, we generated type-B RR11-overexpressing plants (Supplementary Fig. 13a) and examined their chilling phenotype and ZmDREB1 gene expression. Overexpression of ZmRR1-OE impaired maize chilling tolerance (Supplementary Fig. 13b,c). In contrast, the cold-induced expression of ZmDREB1 in these transgenic plants was either upregulated or not altered (Supplementary Fig. 13d), which cannot explain the chilling sensitive phenotype of ZmRR1-OE plants. Considering the previous finding that expression of DREB1s was not affected by cytokinin signaling under cold stress in Arabidopsis (Jeon et al., 2010), we suggest that ZmRR1 regulates DREB1 expression possibly in a type-B independent manner. In addition, we also observed that ZmRR1 regulates expression of cell-wall remodeling genes, such as ZmCesAs and ZmCTLs. Similarly, Arabidopsis type-A ARR6 acts as a linkage for cell-wall composition and immune response (Bacete et al., 2020). Thus, type-A RRs plays important roles in controlling plant cell wall composition under biotic and abiotic stresses. As ZmRR1 is not transcription factors, it would be interesting to identify ZmRR1-associated transcription factors that directly regulate these ZmCesA and ZmCTL genes.”*

(2) We compared the previous findings on two-component signaling system in Arabidopsis, rice and maize in Discussion part: *“Cytokinin signaling involves a His-Asp phosphorelay, which is the key mechanism for plant growth and stress response (Ha et al., 2012). The plant type-A RRs have been suggested to play*

important roles in mediating plant stress tolerance (Cortleven et al., 2019). In Arabidopsis and rice, the expression of type-A RRs is rapidly induced by cytokinin as well as various abiotic stresses (Bhaskar et al., 2021; Jeon et al., 2010; Shi et al., 2012; To et al., 2007). Consistently, the constitutive expression of type-A RRs not only negatively regulates cytokinin signaling but also confers plant abiotic stress tolerance. ZmRR1 encoding a maize type-A response regulator, has been shown to be induced by cytokinin and be necessary for negative feedback regulation of cytokinin signaling (Deji et al., 2000; Hirose et al., 2003). In this study, we demonstrated that ZmRR1 plays a crucial role in enhancing maize cold tolerance. Notably the accumulation of ZmRR1 protein but not the transcription is induced by cold stress. These observations suggest that cold may regulate ZmRR1 at post-translational level.”

(3) We provided the chilling phenotypes of *ZmDREB1.10* overexpression lines to show that ZmDREB1 protein play a positive role in plant response to cold stress (Fig. 6).

Reviewer #3 (Remarks to the Author):

In this manuscript, Zeng and colleagues search for regulators of chilling tolerance in maize. They identify ZmMPK8 as a negative regulator of chilling tolerance that interacts with ZmRR1 and phosphorylates it. They connect ZmRR1 to the expression of chilling responsive genes. This is a well written paper and contains elegant experiments, but the final bioinformatic analysis of the response to cold is disappointing and should be improved.

(1) The authors speculate that the ZmMPK8 repression of ZmRR1 has evolved in order to avoid excess activation of the cold stress pathway. If this is the case, mutants in *zmpk8* and the unphosphorylatable versions of ZmMPK8 which are more cold tolerant should have a yield penalty. Is this the case?

Response: We thank the reviewer for pointing this out. As reviewer mentioned, the crop yield is the most important issue that should be considered. In response to the concern of the reviewer, we provided the yield-related data of *ZmRR1*-overexpression

lines. Overexpressing of *ZmRR1* led to moderately reduced plant height (Supplementary Fig. 12a,b), which could improve lodging-resistance in high-yielding maize. Moreover, few defects in kernel phenotypes, including hundred-grain weight, ear height, ear rows, ear length and ear number, were observed in *ZmRR1*-overexpressing lines under optimal feeding conditions (Supplementary Fig. 12a,c). In addition, we also have measured the hundred-grain weight of wild type and *zmpk8* seeds for three years, and found that there was no significant decrease in *zmpk8* (Response Figure 1). These results suggest that *zmpk8* and the unphosphorylatable versions of ZmMPK8 may not have obvious yield penalty under optimal feeding conditions. Therefore, we delete this speculation in the new version.

Response Fig. 2. The hundred-grain weight of wild-type and *zmpk8* seedlings.

Maize was planted in Zhuozhou, China, in 2018, Sanya, China, in 2019 and 2020.

Data are the mean values \pm SD of three biological replicates.

(2) The labelling of Fig. 4 is very poor. What does CK refer to? Presumably wild-type, but what does CK mean? It says “cold treatment” but doesn’t say the temperature.

Response: We thank the reviewer for the comments. As suggested, we have revised all the labels of Heatmap in the new version of Figure 5 and provided correct and clear information of the experiments for RNA-seq analysis.

(3) Since the data is already there, it seems it would be quite easy to show whether or not there is a systematic alteration of the cold responsive transcriptome. This isn’t shown here. It is not clear for example if there is a control for the 12 h treatment. 0 h is not a good control because lots of circadian and diurnal genes will be differentially

regulated even at 12 h of non-cold temperature. If it could be shown that the team are able to identify the cold responsive transcriptome (even if they have to use another paper to define these genes), then they should show how these 100 or so genes are differentially expressed in their conditions with these genotypes, that would be very helpful.

Response: We thank the reviewer for these good suggestions. In the revised version, we have used published cold responsive transcriptome data (Zhang et al., Plant Cell, 2017) to define ZmRR1-regulated *COR* genes. The results have been described as follows: *Based on the defined cold-responsive transcriptome (Zhang et al., 2017), up to 803 genes is significantly altered by cold treatment, which are termed as ZmRR1-regulated COR genes (Fig. 5c, Supplementary Fig 10).* In addition, we have re-examined the time-course expression of *ZmDREB1* and *ZmCesA* family to confirm their expression is specifically induced by cold (Supplementary Fig 11a).

(4) In Fig. 4 it would make sense to actually have a comprehensive analysis of the RNA-seq data. It seems the reporter assays (e-j) would be better in a separate figure or in the supplemental material. As it is presented, the RNA-seq data is not clear. Similarly, the GO-term enrichments bar chart could also go to the supplementary material. Given that the RNA-seq has been performed the authors should show whether or not it indicates a meaningful mis-regulation of the cold regulon in the different genetic backgrounds.

Response: We thank the reviewer for this good suggestion. We have modified the new Figure 5 as follows: (1) reporter assays have been deleted as reviewer #1 suggested. (2) GO-term enrichments have been replaced to Supplementary Fig10. (3) We overlapped ZmRR1-regulated genes with the defined cold responsive genes from published data (Zhang et al., Plant Cell, 2017), we totally identified 803 ZmRR1-regulated cold regulon (Fig 5 and Supplementary Fig 10).

(5) In Fig. 4b the legend states: heat map showing the log ratios of DEGs, but each column is for one experimental condition. There is no indication of a ratio. For

example it says CK_0h. What is CK? How is this referring to a ratio? A ratio requires two values to be compared, and yet the descriptors refer to individual samples.

Response: We thank the reviewer to point this out. In our revised manuscript, these mistakes have been corrected and shown in Figure 5b legend as follows: *Hierarchical clustering and heat map of 1727 ZmRR1-regulated DEGs and Key GO Terms. DEGs show segregation into 10 co-expressed clusters. The z-score scale represents mean-subtracted regularized log-transformed FPKM.*

(6) In line 290 it says “Heatmap and cluster analyses indicated...” I cannot see any cluster analysis. This is an excellent idea, and I do think the clustering analysis should be performed and presented.

Response: In the revised version, we performed clustering analysis with ZmRR1-regulated genes in Fig 4c. We have described this result as follows: *Gene Ontology (GO) term analysis revealed these ZmRR1-regulated genes significant enriched into 10 gene clusters (<http://bioinfo.cau.edu.cn/agriGO/>). Functional annotation of individual clusters were associated with protein phosphorylation (cluster 1), hydrolase activity (cluster 2), oxidation-reduction (cluster 4), cellulose and cell wall metabolic process (cluster 7 and 8).*

(7) Figure 4 d is presumably using a ratio, but the scale bar is very confusing. The colour scheme is very hard to follow, as is the range. It appears to show that ZmRR1-OE has almost no effect compared to WT on the cold response. This is why a much more comprehensive analysis as described in (3) above is necessary.

Response: We apologize for the confusion. In our revised manuscript, we have unified the scale bar of heatmap to reflect the log₂-fold change comparing relative expression in WT 4°C vs 25°C, *ZmRR1-OE* 4°C vs 25°C and *ZmRR1-OE* 4°C vs WT 4°C (Fig. 5c-d).

(8) The columns in Fig 5a are not labelled in the fig or legend.

Response: As suggested, we have labeled the legend on Fig 5a (which is presented in Fig 5c in the revised version).

(9) The same description is used for Fig 4b,4d and 5a ('log ratios') but the scale bar in Fig 4d is different from Fig 4b,5a. And column names for these 3 figures are totally different (4b, no 'vs'; 4d, 'vs'; 5a, nothing).

Response: We thank the reviewer for pointing out these mistakes. In our revised manuscript, these mistakes have been corrected and shown in Fig 5b-c.

REVIEWERS' COMMENTS

Reviewer #1 (Remarks to the Author):

The authors have answered all my questions.
I do not have further questions.

I have been asked by the editor to comment on the responses to Reviewer #3's previously suggestions.
I think that the authors have solved all the issues.

Reviewer #2 (Remarks to the Author):

The authors properly addressed the comments.

Responses to comments by Editor and Reviewers

Your manuscript entitled "Natural variation in a type-A response regulator confers maize chilling tolerance" has now been seen again by two of our previous referees, whose comments appear below. In light of their advice I am delighted to say that we are happy, in principle, to publish a suitably revised version in Nature Communications under the open access CC BY license.

In particular, we noted the data presented in Figure 3h and 3i are controversial. If MPK8 acts by phosphorylating RR1, you would expect no difference in chilling tolerance if you knock out MPK8 in a *rr1* null mutant background. According to Figure 3h, *zmrr-c1* and *zmmpk8-c1xmrr1-c1* displayed no statistically difference in terms of injured area (%), which is as expected. However, in Figure 3i, the ion leakage (%) difference between the two lines was significant. Editorially, we ask you to tone down the related statements in both the related text of the article and the legend of Figure 7. You can say that MPK8 and RR regulate chilling tolerance, and the MPK8 can phosphorylate RR1, but please avoid to say that MPK8 acts upstream of RR1 or that MPK8 phosphorylates RR1 thereby negatively regulates chilling tolerance.

Response: Thanks for this valuable suggestion. In the last version, the data in Figure 3i represent the mean \pm SEM of three independent experiments (each experiment has three technical repeats. According to the data presentation required for NC (see detailed information as below), as SEM is shorter than SD, it is not encouraged. Therefore, we redone all statistic analyses of the whole ms, and replaced s.e.m. with s.d.. From the new data presented, the ion leakage (%) difference between *zmrr-c1* and *zmmpk8-c1xmrr1-c1* was not statistically significant. Even in this case, we toned down the related statements and rephrased the sentences 'MPK8 acts upstream of RR1 and that MPK8 phosphorylates RR1 thereby negatively regulates chilling tolerance' accordingly.

of data points can lead to the same mean and variability. Moreover, providing only statistical parameters (such as mean \pm s.d. or mean \pm s.e.m., and number of samples) can suggest that the data underlying any particular bar are normally distributed and contains no outliers, when this may not be the case. Graphing error bars with the s.e.m. (which indicates the precision of the mean) is commonly done because they are shorter than error bars representing the s.d. (which instead quantifies variability). We discourage this practice.